# MAS-Zero: Designing Multi-Agent Systems with Zero Supervision

## Abstract

Multi-agent systems (MAS) leveraging the impressive capabilities of Large Language Models (LLMs) hold significant potential for tackling complex tasks. However, most current MAS depend on manually designed agent roles and communication protocols. These manual designs often fail to align with the underlying LLMs' strengths and struggle to adapt to novel tasks. Recent automatic MAS approaches attempt to mitigate these limitations but typically necessitate a validation set for tuning and yield static MAS designs lacking adaptability during inference, while also removing the flexibility to reduce to simpler systems. We introduce MAS-Zero, the first self-evolved, inference-time framework for automatic MAS design. MAS-Zero employs meta-level design to iteratively design, critique, and refine MAS configurations tailored to *each problem instance*, without requiring a validation set. Critically, it enables dynamic problem decomposition and agent composition through meta-feedback on solvability and completeness, and reduction to simpler systems when appropriate. Experiments across reasoning (math and graduate-level QA), coding, and agentic (search-based) benchmarks, using both closed-source and open-source LLM backbones of varying sizes, demonstrate that MAS-Zero outperforms strong manual and automatic MAS baselines. It achieves substantial average accuracy improvements of up to 16.69% on reasoning, 16.66% on coding, and 5.45% on agentic tasks, while maintaining cost efficiency.

## 1 Introduction

While standalone large language models (LLMs) have demonstrated strong performance across numerous tasks (DeepSeek-AI, 2025; Ke et al., 2025b; Vu et al., 2024), many problems remain too intricate for a single model to solve effectively (Wang et al., 2024b; Guo et al., 2024). To tackle these challenges, the exploration of multi-agent systems (MAS) composed of multiple LLM agents has gained increasing traction among researchers (Ke et al., 2025a).[1] These agents often assume distinct *roles*, such as generator or verifier (Shinn et al., 2024), engage in debates offering varied perspectives (Qian et al., 2025; Wang et al., 2024a), and perform assigned sub-tasks (Li et al., 2025).

A fundamental challenge in MAS lies in designing an effective connection and configuration of these agents to solve a given problem. Initially, MAS were handcrafted, with humans designing both agent roles and inter-agent communication protocols. However, MAS composed entirely of such manually designed configurations have faced issues such as poor problem specification and inter-agent misalignment (Cemri et al., 2025; Qiao et al., 2025), especially when the MAS agents are not specifically trained with such configurations.

These shortcomings are understandable, as manually specifying a MAS is difficult when the human designer and the underlying LLMs are not well aligned. Moreover, manual approaches do not scale well to novel problems, especially as the problems become more complex. Recent work has explored automatic MAS design, but they have significant limitations: (1) Most rely on a "training" phase with labeled validation sets to tune configurations, which are often unavailable in real-world scenarios and may not generalize. This training, based solely on *outcome* correctness, provides limited insight into

---

[1]Agents in a MAS can interact with external environmental tools e.g., search tools (Jiang et al., 2024), or collaborate with other agents to address tasks requiring diverse capabilities or multiple steps (Liang et al., 2023; Chen et al., 2024b). This work focuses on the latter scenario, where each agent within the MAS is an LLM communicating with other LLM agents.

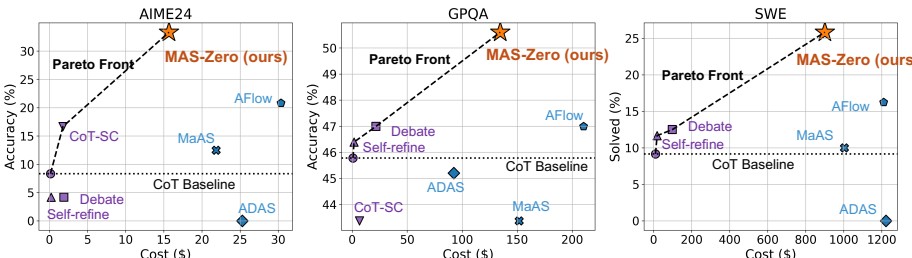

Figure 1: Scatter plots comparing the Pareto fronts of various GPT-4o-based systems on three benchmarks, including manual MAS , automatic MAS and MAS-ZERO . MAS-ZERO delivers high performance at lower cost than comparable automatic MAS methods, establishing a new frontier for accuracy vs. cost trade-off.

the system's internal dynamics. (2) This reliance on validation sets often yields a *fixed* architecture (i.e., one for the entire problem set) which lacks per-problem adaptability at test time. (3) Even worse, these methods eliminate critical dynamics: they cannot reduce to a simpler MAS or a single-agent system when such strategies would be stronger (Huang et al., 2024), nor can they flexibly decompose a problem into smaller, more manageable sub-tasks. This limitation is less apparent on simple tasks such as GSM8K and HumanEval, which are commonly used for MAS evaluation (Hu et al., 2025b; Zhang et al., 2025a;c) and where most methods already perform well. On more challenging tasks (e.g., AIME24), however, the inability to revert or decompose becomes critical: as shown in Fig. 1, many baselines show little to no improvement over simple CoT, meaning the integrated system does not even outperform a single component.

To overcome these limitations, we argue that an effective automatic MAS should satisfy three core desiderata: **(1)** be dynamic enough to both decompose complex problems into smaller sub-tasks and reduce to a single agent or a simple MAS when a sophisticated MAS is *not needed*; **(2)** learn the capabilities of the underlying LLMs, and automatically design MAS structures that are aligned with those capabilities; and **(3)** support adaptivity at inference time, so that MAS designs can be tailored *per problem instance* without relying on training or validation sets. To our knowledge, no existing automatic MAS framework satisfies all three desiderata simultaneously. In this work, we propose a novel automatic inference-time MAS optimization framework, called ***MAS-ZERO***, which designs MAS with *zero* supervision, while satisfying all the aforementioned desiderata. In particular, MAS-ZERO introduces a **meta-agent** that iteratively learns the capabilities of individual agents and their combinations, and refines the MAS design accordingly, thus operating at the MAS-level rather than the agent level (hence "meta"). The meta-agent also verifies candidate answers drawn from both refined MAS designs and simpler MAS or single-agent systems, ensuring the dynamic reduction capability. This process operates entirely at test time, allowing for unique MAS designs *per-problem*.

To achieve this, MAS-ZERO tasks the meta-agent to iteratively *design* and *critique* the MAS, maintain an *experience library*, *refine* the design based on accumulated experience, and ultimately *verify* the candidate answers. Fig. 2 illustrates a conceptual overview and contrasts MAS-ZERO with both automatic and manual MAS designs. Specifically, MAS-ZERO involves three key steps:

- **Initializing building blocks (MAS-Init)**: MAS-ZERO starts with established single-agent (e.g., CoT, Self-Consistency) and simple human-designed MAS strategies (e.g., Debate, Self-Refine), executing each to generate initial outputs that seed later steps.
- **Self-evolving with iterative refinement (MAS-Evolve)**: The meta-agent iteratively *designs* and *critiques* MAS configurations, guided by feedback on *solvability* and *completeness*, while accumulating prior designs and feedback in an experience library for continual refinement.
- **Selecting the best candidate with self-verification (MAS-Verify)**: From the pool of outputs, including both building blocks and refined MAS iterations, the meta-agent verifies and selects the most reliable solution, dynamically choosing between complex MAS and simpler strategies.

Evaluations across three challenging domains—reasoning (math and graduate-level QA), coding, and agentic (search-based), using both closed-source and open-source LLM backbones of varying sizes (including GPT-4o, 32B, and 70B models) demonstrate that MAS-ZERO consistently outperforms strong manual and automatic MAS baselines. It achieves substantial average accuracy improvements of up to 16.69% on reasoning, 16.66% on coding, and 5.45% on agentic tasks. It also consistently lies on the Pareto frontier of accuracy and cost (Fig. 1). While the inference-time mechanism incurs

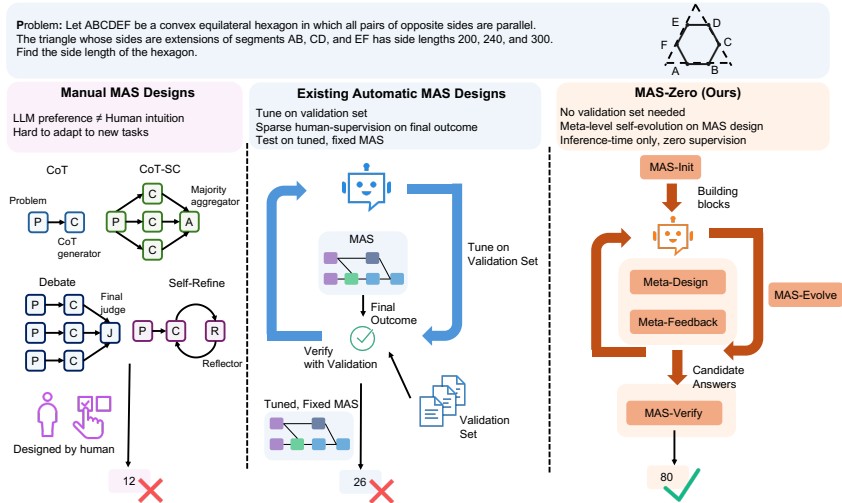

Figure 2: Conceptual comparison of MAS-ZERO, with existing automatic and manual MAS designs. MAS-ZERO avoids tuning MAS on validation set by maintaining a self-evolving process that iteratively designs and evaluates task-specific MAS at inference time.

higher token usage during testing, it avoids expensive validation-time optimization and shifts the design effort to the testing phase, where it can flexibly handle new tasks, and often be more effective (Agrawal et al., 2025). Such a trade-off has demonstrated significantly improved answers in this work and strong potential in the literature (Liu et al., 2024a). We believe that MAS-ZERO provides a complementary alternative for the MAS community, especially in scenarios where adaptability and generality outweigh the need for minimal token usage. In summary, our key contributions are:

- We introduce MAS-ZERO, to our knowledge, the **first inference-time-only** automatic MAS design framework. It works in a fully self-evolved way by learning from the behavior of the underlying LLM agents *at inference-time*, enabling per-instance adaptivity with *zero* supervision.
- We present **a new SoTA automatic MAS system** that achieves substantial performance gains over both manually designed and strong automatic baselines, while remaining cost-efficient and Pareto-optimal across diverse LLMs and domains.[2]
- We conduct a comprehensive evaluation of MAS-ZERO across diverse domains and LLMs, presenting **key insights**. For example, single-agent or simple MAS configurations can be surprisingly strong, in some cases even outperforming more sophisticated MAS designs. Crucially, MAS-ZERO is the only system that can *dynamically revert* to these simpler yet effective strategies, ensuring that such strengths are not overlooked.

## 2 RELATED WORK

**Manual MAS design.** Building on the success of single-agent systems (e.g., CoT (Wei et al., 2022), self-consistency (CoT-SC) (Wang et al., 2023a)), studies have shown that grouping multiple LLM agents into a MAS can substantially improve individual agent performance. To this end, a variety of manual-designed MAS approaches have been proposed (Xu et al., 2025; Zheng et al., 2024; Lu et al., 2025), including LLM debate (Du et al., 2023), and self-refine (Madaan et al., 2024). However, as discussed previously, these manual designs often suffer from limited adaptability and scalability, and their rigid structures may fail to align with the underlying strengths of LLMs.

**Automatic MAS design.** Recent work on automatic MAS design typically require validation set. We broadly categorize them into two families: **(1) val-pruning** starts with a fully connected, pre-defined graph of LLM agents or human-designed blocks and prune it based on validation performance. For example, MASS (Zhou et al., 2025a) uses rejection sampling, and MaAS (Zhang et al., 2025a) extends MASS with a question-wise masking mechanism to adapt subnetworks. However, their adaptability remains limited as the core MAS structure is constrained by the pre-defined structure, which is suboptimal for many tasks; **(2) val-generation** leverages a meta-agent LLM to generate MAS from

---

[2]We will open-source the data, code, and leaderboard for all components upon acceptance.

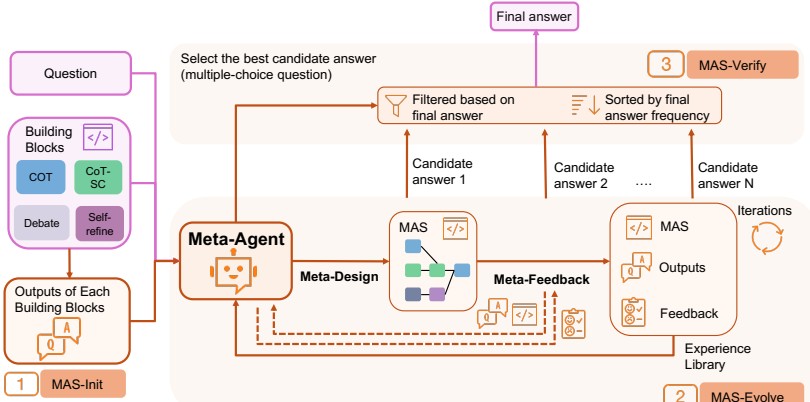

Figure 3: MAS-ZERO overview. Purple highlights the given input and final output. Orange highlights the components and steps in MAS-ZERO. Dashed arrows indicate the information flow *within* Meta-feedback. MAS-ZERO takes as inputs the question and building blocks, and solves the task in three key steps: **MAS-Init** (Sec. 3.1), **MAS-Evolve** (Sec. 3.2) and **MAS-Verify** (Sec. 3.3).

scratch, offering greater flexibility in defining novel agents and architectures compared to pre-defined structures. However, this expanded design space presents significant learning challenges. Recent efforts including ADAS (Hu et al., 2025b) and AFlow (Zhang et al., 2025c) frame MAS generation as a code generation task. ADAS stores and searches historical designs based on validation performance, while AFlow enhances this with Monte Carlo Tree Search. Our framework also represents MAS as executable code but differs fundamentally in several ways: instead of relying on potentially unreliable validation sets, MAS-ZERO uses a self-evolving process at inference time to learn the capabilities of agents for meta-level design. It further integrates question decomposition into MAS design, enabling MAS to be constructed and refined at the sub-task level. Finally, MAS-ZERO can dynamically revert to simpler building blocks when they are sufficient. These capabilities are not supported by existing automatic MAS systems. More methods like DyLAN (Liu et al., 2024b) are discussed in App. D.

## 3 MAS-ZERO FRAMEWORK

As shown in Fig. 3, MAS-ZERO first conducts **MAS-Init** (Sec. 3.1), where it executes each of the given building blocks. It then takes a question, the building blocks, and the outputs of each building block as inputs, ultimately producing the final answer. These inputs are processed by the central **meta-agent**, which orchestrates both the **MAS-Evolve** (Sec. 3.2) and **MAS-Verify** (Sec. 3.3) steps. Importantly, the whole process functions *without* prior knowledge or internal details of the underlying LLM agents. All steps are implemented through prompting and require only black-box access to LLM generation, making MAS-ZERO broadly applicable to any LLM without requiring fine-tuning or internal modifications. The corresponding pseudocode is provided in App. A.

### 3.1 MAS-INIT

MAS-Init serves as the entry point of MAS-ZERO by executing a set of predefined *building blocks*. These blocks correspond to established human-designed strategies (CoT, CoT-SC, Debate, and Self-Refine in this work) implemented as executable code. Given a question, MAS-Init runs each block to generate initial candidate solutions. These blocks and their outputs are used as: (1) input to the meta-agent for grounding the MAS design (Sec. 3.2), and (2) candidate answers that can be selected by the MAS-Verify (Sec. 3.3), enabling dynamic reduction to simpler MAS or single-agent systems.

### 3.2 MAS-EVOLVE

Given the question, the building blocks, and their outputs from MAS-Init, the meta-agent begins to design the MAS. Initially, it has no knowledge of the underlying LLM agents' internal capabilities and may produce suboptimal designs. We propose an iterative process in which the meta-agent gradually learns the strengths of the component agents and refines its designs. This process alternates between two phases: **(1) meta-design** (Sec. 3.2.1), where the meta-agent decomposes the question into sub-tasks and proposes a MAS based on the building blocks and any accumulated *experience* from prior

iterations; **(2) meta-feedback** (Sec. 3.2.2), where the meta-agent reviews the proposed MAS and sub-tasks using intermediate outputs to assess their solvability and completeness, and then generates targeted feedback. The MAS, its intermediate outputs and feedback, are stored in an *experience library* that informs subsequent iterations. Through this cycle, the meta-agent progressively adjusts decompositions and configurations, yielding continual improvement without external supervision.

### 3.2.1 META-DESIGN

The goal of this phase is to design a candidate MAS for the given task, which will then be reviewed in the next phase. Unlike existing work that tackles complex problems all at once, MAS-ZERO explicitly decomposes the original question into *manageable* yet *interdependent* sub-tasks. This decomposition not only breaks down complex problems into smaller parts but also creates opportunities to assign sub-task level MAS (i.e., *sub-MAS*) tailored to different components of the problem. For each sub-task, the meta-agent assigns a sub-MAS by modifying connections between given building blocks or adjusting their parameters (e.g., temperature, number of debate rounds, etc.). This deliberate design, informed by our preliminary experiments, balances exploration with improvement: the meta-agent is free to analyze questions, sub-tasks and assigned sub-MAS, but it should not arbitrarily invent new agents or blocks, nor prune the architecture without grounding in the provided building blocks.[3]

### 3.2.2 META-FEEDBACK

**MAS and intermediate outputs.** Given the design produced in the meta-design phase, the meta-feedback phase reviews the MAS and generates feedback. Since a MAS is executable code, it can be run to obtain outputs, but relying only on the final answer is often sparse and uninformative. MAS-ZERO instead exploits the intermediate outputs, incorporating both *sub-task level* outputs from sub-MAS and *agent-level* outputs from individual LLMs. By jointly analyzing the final answer and these fine-grained signals, the meta-agent gains a much richer view of strengths and weaknesses across the MAS. Concretely, with the code-based representation, each sub-MAS is executed to solve its sub-task, producing intermediate outputs at two levels: the *sub-task (sub-MAS)* level and the *agent* level. These outputs form the basis for evaluation against the key criteria introduced below.

**Criteria.** Given the above sub-task and agent level outputs, MAS-ZERO evaluates *solvability* and *completeness*. The meta-agent is given agency in determining each metric:

- **Solvability** requires that each sub-task be *independently* and *completely* solvable by its sub-MAS, ensuring that every sub-task yields reliable outputs.[4]
- **Completeness** requires that the complete set of sub-tasks covers all necessary information from the original input, ensuring that their answers can produce a correct and comprehensive aggregated answer to the original task. While an individual sub-task may address only part of the necessary content, all critical information must be processed and used at some point in the MAS.

**Generating feedback.** Based on the solvability and completeness, the meta-agent generates targeted natural language feedback on specific aspects of the MAS that may require revision. For example, if a sub-task is identified as not solvable, the feedback should suggest either further decomposing it or updating the corresponding sub-MAS in the next iteration. Conversely, if a sub-task is considered solvable, the feedback should indicate that it and its sub-MAS remain unchanged. Similarly, if the union of sub-tasks is found to miss necessary information, the feedback should recommend refining the decomposition of the original problem to incorporate the missing elements. Overall, this feedback guides subsequent meta-design iterations, allowing the overall system to iteratively converge toward an effective decomposition and MAS.

### 3.2.3 STORING THE EXPERIENCE AND REFINING THE DESIGN

After the first meta-design (Sec. 3.2.1), meta-feedback is collected (Sec. 3.2.2). The MAS, its intermediate outputs, and the associated feedback are stored as *experience* in an *experience library*. In each subsequent iteration, meta-design is performed again, now with experience from the library provided

---

[3]To support code generation, we provide a template with utility functions and apply sanity checks (syntax validation and field consistency; see App. I).

[4]To further aid the meta-agent, we allow each agent to output a special token, [TOO HARD], if it determines that the assigned sub-task is beyond its current capabilities.

as additional context to drive self-evolution. Through this process, the meta-agent dynamically adapts its decomposition strategy and sub-MAS assignments across iterations. This iterative accumulation of experience gives MAS-ZERO a persistent memory, enabling it to leverage knowledge from past iterations and build a stronger foundation for continual improvement.

As in many other self-evolving frameworks (Gao et al., 2025b), the meta-design and meta-feedback signals may be imperfect and ultimately depend on the underlying LLM. Nevertheless, we find empirically that MAS-ZERO allows initially imperfect designs to be progressively improved (Fig. G.1), and that our curated instruction design produces strong feedback—outperforming a simple ensemble alternative (Sec. 4.2). We view these results as a promising starting point and hope they inspire further research in advancing strategies for iterative MAS refinement

### 3.3 MAS-VERIFY

**Collecting candidate answers.** At each iteration of MAS-Evolve, the MAS is executed to produce intermediate outputs and a *candidate answer* (including both the chain-of-thought and the final answer). After multiple rounds, MAS-ZERO must determine which candidate answer is the most reliable and complete. Importantly, the pool of candidate answers includes *not only* those generated in each iteration of MAS-Evolve *but also* the outputs of the basic building blocks from MAS-Init. This design allows the meta-agent to select between them, leveraging the strong performance of simple strategies when they suffice, while also exploiting the complex MAS when needed.

**Verifying answers.** Relying on the last iteration (or any single iteration) is suboptimal due to stochastic LLM outputs and ongoing MAS refinement (ablations in Sec. 4.2). Instead, MAS-ZERO formulates verification as a *selection* problem and tasks the meta-agent with selecting the most coherent and correct output from the set of candidate answers, which is often more tractable than independently scoring each output (Gu et al., 2025; Zhou et al., 2025b), especially for challenging questions where correctness is hard to assess in isolation. Specifically, MAS-ZERO first *ranks* candidates by their final answer frequency. This acts as a prior favoring majority responses, a strategy shown to be effective in prior work (Wang et al., 2023a). It then *filters* out clearly invalid answers (e.g., not among the given options). Finally, it *selects* the best answer from the remaining candidates.

## 4 EXPERIMENTS

**Setup.** We consider both the closed-source **GPT-4o** (OpenAI, 2023) (web-search version for agentic tasks) and the open-source LLMs, **Llama3.3-70B-inst** (Llama, 2024) and **Qwen2.5-32B-inst** (Qwen, 2025). To fairly evaluate how well MAS-ZERO performs relative to the underlying LLM used to construct the MAS, we always use the *same* LLM for both the meta-agent and individual agents (heterogeneous settings in Sec. 4.2). We use the same prompt template for all the tasks (App. H and I) and conduct 5 MAS-Evolve iterations (the maximum permitted by context-length). Together with the 4 building blocks in MAS-Init, this yields 9 candidate answers, from which the meta-agent selects one final answer with MAS-Verify.

**Benchmarks.** We consider 2 **reasoning** benchmarks across math and science: **AIME24** (AIME, 2024) and GPQA-diamond **(GPQA)** (Rein et al., 2023) (graduate-level QA), 1 **coding** benchmark SWE-Bench-Lite-Oracle **(SWE)** (Jimenez et al., 2024),[5] and 2 search-based **agentic** benchmarks: **BrowseComp** (Wei et al., 2025) and **Frames** (Krishna et al., 2025).[6] Existing automatic MAS methods largely restrict their evaluations to relatively simple reasoning tasks. To our knowledge, MAS-ZERO is the first to conduct evaluations on challenging reasoning, coding and agentic tasks.

**Baselines.** We include 2 widely used **single-agent** baselines: **CoT** (Wei et al., 2022) and self-consistency **(CoT-SC)** (Wang et al., 2023a); 6 **manual MAS** baselines: **Debate** (Du et al., 2023), **Self-refine** (Madaan et al., 2024), **ReConcile** (Chen et al., 2024b), **MAD** (Liang et al., 2023), **SPP** (Wang et al., 2023b) and **DyLAN** (Liu et al., 2024b). Note that CoT, CoT-SC, Debate and Self-Refine also serve as the building blocks in MAS-Init, allowing us to clearly observe how our system improves upon the initial configurations. For **automatic MAS**, we include 3 strong methods: **val-pruning MaAS** (Zhang et al., 2025a) and **val-generation ADAS** (Hu et al., 2025b) and **AFlow** (Zhang et al., 2025c). We also include the latest **training-based** method **MAS-GPT** (Ye et al., 2025).

---

[5] Note that MAS-Verify does not apply to SWE, as correctness in SWE is determined directly by the compiler.
[6] Benchmark statistics and more implementation details can be found in App. B.

| LLMs | GPT-4o | | | Llama3.3-70B | | | Qwen2.5-32B | | |
|---|---|---|---|---|---|---|---|---|---|
| Methods | AIME24 | GPQA | Avg. | AIME24 | GPQA | Avg. | AIME24 | GPQA | Avg. |
| CoT | 8.33 | 45.78 | $27.06_{\uparrow 14.91}$ | 16.67 | 50.60 | $33.63_{\uparrow 11.32}$ | 12.50 | 50.00 | $31.25_{\uparrow 9.24}$ |
| CoT-SC | 16.67 | 43.37 | $30.02_{\uparrow 11.95}$ | 29.17 | 51.20 | $40.18_{\uparrow 4.77}$ | 16.67 | 49.40 | $33.04_{\uparrow 7.46}$ |
| Debate | 4.17 | 46.99 | $25.58_{\uparrow 16.39}$ | 20.83 | 50.60 | $35.72_{\uparrow 9.24}$ | 8.33 | 49.40 | $28.87_{\uparrow 11.63}$ |
| Self-Refine | 4.17 | 46.39 | $25.28_{\uparrow 16.69}$ | 29.17 | 54.22 | $41.69_{\uparrow 3.26}$ | 16.67 | 50.60 | $33.64_{\uparrow 6.86}$ |
| ReConcile | 12.50 | 48.43 | $30.47_{\uparrow 11.50}$ | 33.33 | 47.17 | $40.25_{\uparrow 4.71}$ | 12.50 | 47.17 | $29.84_{\uparrow 10.66}$ |
| MAD | 13.89 | 52.01 | $32.95_{\uparrow 9.02}$ | 29.17 | 52.61 | $40.89_{\uparrow 4.07}$ | 16.67 | 43.57 | $30.12_{\uparrow 10.37}$ |
| SPP | 9.72 | 49.80 | $29.76_{\uparrow 12.21}$ | 26.39 | 46.18 | $36.29_{\uparrow 8.67}$ | 19.44 | 42.77 | $31.11_{\uparrow 9.39}$ |
| DyLAN | 11.11 | 46.99 | $29.05_{\uparrow 12.92}$ | 29.17 | 41.57 | $35.37_{\uparrow 9.59}$ | 20.83 | 42.57 | $31.70_{\uparrow 8.79}$ |
| MaAS | 12.50 | 43.37 | $27.94_{\uparrow 14.03}$ | 33.33 | 43.98 | $38.65_{\uparrow 6.30}$ | 20.83 | 46.99 | $33.91_{\uparrow 6.58}$ |
| ADAS | × | 45.20 | × | 8.30 | 53.60 | $30.95_{\uparrow 14.01}$ | 12.50 | 47.00 | $29.75_{\uparrow 10.74}$ |
| AFlow | 20.83 | 46.99 | $33.91_{\uparrow 8.05}$ | 33.33 | 47.59 | $40.46_{\uparrow 4.49}$ | 33.33 | 46.39 | $39.86_{\uparrow 0.63}$ |
| MAS-GPT | 13.89 | 43.98 | $28.94_{\uparrow 13.03}$ | 26.39 | 40.00 | $33.20_{\uparrow 11.76}$ | 23.61 | 37.35 | $30.48_{\uparrow 10.01}$ |
| MAS-ZERO | 33.33 | 50.60 | **41.97** | 37.50 | 52.41 | **44.96** | 29.17 | 51.81 | **40.49** |

Table 1: Reasoning tasks results. "×" indicates 0% accuracy for MAS selected using the validation set. "↑" denotes the difference (improvement) that MAS-ZERO achieves compared to the baselines. Highlighting indicates single-agent, manual MAS, val-pruning automatic MAS, val-generation automatic MAS, training-based automatic MAS and our method. To fairly compare with validation-based baselines, we split each benchmark's original test set into 20% for validation and 80% for testing. For methods do not use validation sets (including MAS-ZERO), we evaluate on the same 80% split. Standard deviations are given in App. E.

## 4.1 OVERALL RESULTS

Tables 1-3 show the results for reasoning, coding and agentic tasks across applicable LLMs and benchmarks. On average, MAS-ZERO achieves the *best* performance across all LLMs and domains. Below, we summarize the additional takeaways from the comparison:

**Reasoning Tasks.** From Table 1, we observe that (1) MAS-ZERO *consistently outperforms* all *automatic MAS* methods. Across all LLM backbones and benchmarks, it surpasses SoTA baselines, exceeding the strongest baseline, AFlow, by 13.03% on average with GPT-4o as the backbone. The only instance where it falls behind is on AIME24 with the Qwen backbone, where it underperforms AFlow by merely one sample (out of 24 total). Notably, ADAS fails completely on AIME24 (0% accuracy), despite having access to a validation set, underscoring the unreliability of validation-based baselines. (2) MAS-ZERO also *consistently outperforms* strong single-agent and manual MAS, with only two exceptions: GPQA with MAD using GPT-4o, and GPQA with Self-Refine using Llama.

Alarmingly, several automatic MAS baselines underperform manual MAS across multiple benchmarks. For example, CoT and CoT-SC consistently outperform MAS-GPT, ADAS, and MaAS. This further highlights the necessity of MAS-Init in MAS-ZERO, as it allows the system to dynamically revert to simpler strategies when a sophisticated MAS is not needed.

**Coding Tasks.** Similar to reasoning tasks, in Table 2 we observe MAS-ZERO clearly outperforms single-agent, manual and automatic MAS. Notably, it comes with 58% (GPT-4o) and 149% (Llama) relative gains over the strongest baseline AFlow. These margins exceed those observed in reasoning tasks, highlighting the effectiveness of MAS-ZERO in challenging tasks.

| LLMs Methods | GPT-4o SWE | Llama3.3 SWE |
|---|---|---|
| CoT | $9.17_{\uparrow 16.66}$ | $2.92_{\uparrow 13.82}$ |
| Debate | $12.50_{\uparrow 13.33}$ | $6.67_{\uparrow 10.07}$ |
| Self-Refine | $11.67_{\uparrow 14.16}$ | $1.67_{\uparrow 15.07}$ |
| MaAS | $10.00_{\uparrow 15.83}$ | $5.00_{\uparrow 11.74}$ |
| AFlow | $16.25_{\uparrow 9.58}$ | $6.67_{\uparrow 10.07}$ |
| MAS-ZERO | **25.83** | **16.74** |

Table 2: SWE results. Methods that cannot be adapted to SWE are not included. Qwen is not included due to its small maximum context length (32K).

**Agentic Tasks.** We use GPT-4o with search as individual agent, which can query the internet and conduct multi-turn autonomous reasoning internally (meta-agent is still GPT-4o). From Table 3, we see that on average, MAS-ZERO continues to improve upon the basic building blocks. On Frames, MAS-ZERO underperforms Debate and Self-Refine. We speculate that when the search agent makes mistakes, those errors are grounded in retrieved content, making them more difficult to detect during MAS-Verify, leading to incorrect judgments. This highlights the importance of further strengthening the verifier in MAS-Verify (see Sec. 4.2 for more analysis).

| LLM Methods | GPT-4o w/ search | | |
|---|---|---|---|
| | BrowseComp | Frames | Avg. |
| CoT | 3.97 | 59.76 | $31.86_{\uparrow 5.45}$ |
| CoT-SC | 8.66 | 63.58 | $36.12_{\uparrow 1.19}$ |
| Debate | 3.94 | 70.45 | $37.19_{\uparrow 0.12}$ |
| Self-Refine | 5.51 | 67.89 | $36.70_{\uparrow 0.61}$ |
| MAS-ZERO | 9.45 | 65.18 | **37.31** |

Table 3: Results on agentic tasks.

**Cost-efficiency.** Fig. 1 shows the trade-off between performance and cost for GPT-4o across three benchmarks. Cost is computed using the official OpenAI API pricing[7] and includes both "training" (if any) and test-time usage. We observe that MAS-ZERO **lies on the Pareto front across all three datasets**. It is significantly more cost-efficient than AFlow, MaAS, and ADAS, with the lone exception of ADAS on GPQA, where the cost increase comes with a 12% accuracy improvement. Of automatic MAS frameworks, MAS-ZERO delivers the highest performance at relatively low cost. While it is expected that automatic MAS methods incur higher costs than manual baselines, MAS-ZERO delivers substantially better performance, making the trade-off highly favorable.

## 4.2 FURTHER ANALYSIS AND ABLATIONS

While Sec. 4.1 establishes the overall effectiveness of MAS-ZERO across domains and LLMs, here we analyze the role of the **meta-agent** and **each of the three steps** through a series of targeted ablations. The results, detailed below, show that a capable meta-agent consistently enhances performance and that all three steps contribute meaningfully and complementarily to the final improvements.

**Diverse meta-agents.** While MAS-ZERO shows strong performance across various LLMs, we further examine whether weaker or stronger LLMs can effectively serve as meta-agents. For stronger LLM, we conduct experiments with a reasoning LLM, **o3-mini** (OpenAI, 2025b). As shown in Table 4, MAS-ZERO outperforms the considered baselines on average, indicating that the benefits of MAS-ZERO generalize well across model strengths. For weaker LLMs, we conduct experiments with **GPT-OSS-**

| LLM | o3-mini | | |
|---|---|---|---|
| Methods | AIME24 | GPQA | Avg. |
| CoT | 70.00 | 72.22 | $71.11_{\uparrow 12.27}$ |
| CoT-SC | 80.00 | 72.73 | $76.36_{\uparrow 7.02}$ |
| Debate | 86.67 | 77.78 | $82.22_{\uparrow 1.16}$ |
| Self-Refine | 76.67 | 74.24 | $75.45_{\uparrow 7.93}$ |
| MAS-ZERO | 90.00 | 76.77 | **83.38** |

Table 4: MAS-ZERO with stronger agents.

**20B** (OpenAI et al., 2025), **Qwen2.5-7B** (Qwen, 2025), and **Qwen2.5-Coder-3B** (Hui et al., 2024), **GPT-4.1-nano** (OpenAI, 2025a). We find that these models are unable to reliably follow instructions and often produce syntactically incorrect Python code, suggesting that the meta-agent role requires sufficiently strong capabilities to handle its multiple responsibilities.

**Heterogeneous agents.** The previous experiments use the same LLM for both the meta-agent and the individual agents and already achieved strong results. An intriguing question is whether heterogeneous assignments can yield additional benefits or drawbacks. Specifically, we explore pairing a stronger LLM as the meta-agent with a weaker LLM as the individual agent, and vice versa. As shown

| Agent | Meta-agent | AIME24 | GPQA | Avg. |
|---|---|---|---|---|
| GPT-4o | GPT-4o | 33.33 | 50.60 | 41.97 |
| o3-mini | o3-mini | 90.00 | 76.77 | 83.38 |
| GPT-4o | o3-mini | 36.67 | 60.10 | 48.38 |
| o3-mini | GPT-4o | 83.33 | 73.74 | 78.54 |

Table 5: MAS-ZERO with different models.

in Table 5, when GPT-4o is the individual agent and the meta-agent is replaced with o3-mini, performance improves notably but still falls short of directly using o3-mini for both roles. Conversely, when o3-mini is the individual agent and the meta-agent is replaced with GPT-4o, performance decreases, though it remains better than the setting where GPT-4o is the agent and o3-mini is the meta-agent. These results suggest that while a stronger meta-agent can provide benefits, the overall performance is ultimately constrained by the capability of the individual agent.

**MAS-Init.** Table 1 suggests that building blocks can achieve strong performance in some problems. To quantify their contribution, we ablate MAS-Init by skipping execution of the building blocks in the first step, letting MAS-Verify judge *solely* based on the five candidate solutions produced by the five iterations of MAS-Evolve. As shown in Table 6(**-MAS-Init**), this significantly degrades performance, highlighting the importance of including MAS-Init and the ability of MAS-ZERO to dynamically revert to building blocks.

| LLM | GPT-4o | | |
|---|---|---|---|
| Methods | AIME24 | GPQA | Avg. |
| MAS-ZERO | **33.33** | **50.60** | **41.97** |
| - MAS-Init | 12.50 | 48.43 | $30.46_{\downarrow 11.50}$ |
| - MAS-Evolve | 20.00 | 48.73 | $34.37_{\downarrow 7.60}$ |
|   - meta-design | 20.83 | 45.18 | $33.01_{\downarrow 8.96}$ |
|   - meta-feedback | 25.00 | 42.17 | $33.59_{\downarrow 8.38}$ |
|   → ensemble meta-feedback | 16.67 | 46.88 | $31.77_{\downarrow 10.19}$ |
| - MAS-Verify | 6.70 | 33.83 | $20.27_{\downarrow 21.70}$ |

Table 6: Ablations on the three steps in MAS-ZERO.

**MAS-Evolve.** To evaluate its importance, we first conduct an ablation by skipping the entire MAS-Evolve and letting MAS-Verify judge *solely* based on the four building block outputs from MAS-Init. As shown in Table 6(**-MAS-Evolve**), the performance drops notably, indicating that MAS-Evolve

---

[7]More details are given in App. C

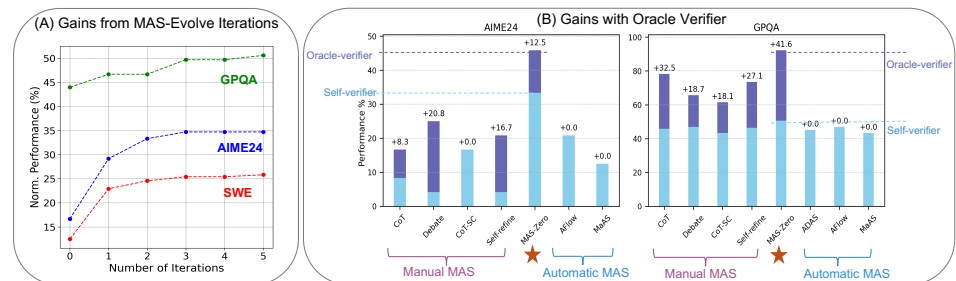

Figure 4: **(A)** Performance gains (GPT-4o) over MAS-Evolve. **(B)** Performance gains (GPT-4o) given an oracle verifier. Automatic MAS baselines cannot integrate external verifiers, yielding zero improvement.

is useful for the overall improvements. For **meta-design**, we modify the prompt to ask the meta-agent to propose a MAS configuration *without* attempting to decompose the question into sub-tasks; Table 6 **(-meta-design)** shows that removing decomposition leads to a significant performance drop, demonstrating that breaking down the task is a meaningful contributor to the effectiveness of MAS-ZERO. For **meta-feedback**, we test two variants: (1) modifying the prompt so that the meta-agent critiques the current MAS *without* analyzing the solvability and completeness of each sub-task or LLM agent **(-meta-feedback)**; (2) since meta-feedback can be noisy due to the self-evolving nature of the system, we explore a straightforward method to improve reliability via ensembling ($\rightarrow$ **ensemble meta-feedback**). Following Du et al. (2023), we generate multiple feedback candidates (three in our experiments) from the meta-agent and then use an additional call to the meta-agent to select the best one. The corresponding rows in Table 6 reveal that removing meta-feedback results in a large performance drop, confirming that it is critical to the overall effectiveness of MAS-ZERO. Surprisingly, the ensemble approach not only fails to improve performance but even reduces it. This counter-intuitive result suggests that the current straightforward meta-feedback is already strong, and advancing MAS-ZERO will require designing more principled strategies for reliable feedback.

**Gains from MAS-Evolve at each iteration.** To further evaluate the self-evolving capability in MAS-Evolve, we examine performance across iterations. As shown in Fig. 4(A), accuracy at iteration 0 (before MAS-Evolve, only MAS-Init) and 1 (after the first refinement) is notably lower, indicating that the system struggles to design effective MAS at the outset. With subsequent iterations, however, MAS-ZERO progressively improves, demonstrating a strong ability to self-evolve through the refinement cycle of meta-design, meta-feedback, and the accumulated experience library.

**MAS-Verify.** This final step determines which candidate solution is selected as the final answer. To assess its importance, we conducted an ablation where the system simply used the *last* iteration as the output, without any additional judgment. The last row in Table 6 shows a sharp performance decline (the largest drop among all ablations). This is intuitive because, as shown earlier, the ability to dynamically revert to building blocks is indispensable. Yet it is also revealing, since the self-evolving nature of MAS-ZERO might suggest that the final iteration should yield the strongest solution. Instead, the outcome highlights that iterative refinement alone is insufficient, and that effective verification is essential to counteract the stochasticity of pure self-evolving methods without ground-truth signals.

**Potential of MAS-Verify with oracle verifier.** We showed that MAS-Verify is crucial, and this highlights substantial headroom for further improvement. Unlike existing automatic MAS frameworks, MAS-ZERO can seamlessly incorporate *external verifiers*, making it naturally positioned to benefit from advances in verification techniques. Fig. 4(B) illustrates this potential: when equipped with an *oracle verifier* that labels outputs as "correct" or "incorrect" using ground-truth answers, MAS-ZERO 's performance improves dramatically, further widening the gap over both manual and automatic MAS and pushing GPQA close to 95%. This demonstrates that stronger verification could unlock significant headroom for MAS-ZERO.

## 5 CONCLUSION

We presented MAS-ZERO, the first *inference-time-only* automatic MAS design framework with *zero* supervision. It iteratively designs and refines MAS, decomposes complex questions, reverts to simpler strategies when sufficient, and verifies candidate answers. Comprehensive experiments show its strong effectiveness, cost-efficiency, and the contribution of each step.

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

## A  MAS-ZERO ALGORITHM

In Section 3, we details the three steps of MAS-ZERO. Algorithm 1 presents the detailed algorithm. Highlighting indicates  MAS-Init ,  MAS-Evolve  and  MAS-Verify .

---

**Algorithm 1:** MAS-ZERO: Designing Multi-Agent Systems with Zero Supervision

---

**Input:** Question $Q$, building blocks $\{\mathcal{M}^{(1)}, \ldots, \mathcal{M}^{(k)}\}$, meta-agent $\mathcal{A}$, iterations $T$
**Output:** Final Answer $y^*$
1 **Initialize** candidate answers $\mathcal{H} \leftarrow [\,]$, experience library $\mathcal{E} \leftarrow \varnothing$;

2  **Step 1: MAS-Init**

3 **foreach** *building block* $\mathcal{M}^{(i)}$ **do**
4     $Y_0^{(i)} \leftarrow \texttt{Execute}(\mathcal{M}^{(i)}, Q)$ ;                            `// Run each building block`
5     extract final answer $y_0^{(i)}$ from $Y_0^{(i)}$;
6     append $y_0^{(i)}$ to $\mathcal{H}$;
7     $\mathcal{E} \leftarrow \mathcal{E} \cup \{(Q, \mathcal{M}^{(i)}, y_0^{(i)})\}$ ;                    `// Store the answers from MAS-Init`

8  **Step 2: MAS-Evolve**

9 $(\mathcal{Q}_0, \mathcal{M}_0) \leftarrow \mathcal{A}.\texttt{Meta\_Design}(Q, \{\mathcal{M}^{(i)}\}, \mathcal{E}, \text{Constraints} = \{\mathcal{M}^{(i)}\})$;
   `// Decompose into sub-tasks` $\mathcal{Q}_0$ `and assign sub-MAS` $\mathcal{M}_0$ `grounded in building blocks`
10 **for** $t = 1$ **to** $T$ **do**
11     $Y_t \leftarrow \texttt{Execute}(\mathcal{M}_{t-1}, \mathcal{Q}_{t-1})$ ;                      `// Run current MAS on sub-tasks`
12     extract sub-task outputs $\{(x_j^{\text{sub}}, y_j^{\text{sub}})\}$ and agent outputs $\{(x_\ell^{\text{agent}}, y_\ell^{\text{agent}})\}$ from $Y_t$;
13     $(\mathcal{Q}_t, \mathcal{M}_t, y_t, f_t) \leftarrow \mathcal{A}.\texttt{Meta\_Feedback}(Q, \mathcal{Q}_{t-1}, \mathcal{M}_{t-1}, \{(x_j^{\text{sub}}, y_j^{\text{sub}})\}, \{(x_\ell^{\text{agent}}, y_\ell^{\text{agent}})\}, \mathcal{E}, \text{Constraints} = \{\mathcal{M}^{(i)}\})$;
      `// Assess solvability and completeness; revise decomposition and sub-MAS`
14     $\mathcal{E} \leftarrow \mathcal{E} \cup \{(\mathcal{Q}_{t-1}, \mathcal{M}_{t-1}, Y_t, f_t)\}$ ;       `// Store sub-tasks, sub-MAS, intermediate outputs, and feedback`
15     **if** $y_t \neq \perp$ **then**
16        append $y_t$ to $\mathcal{H}$

17  **Step 3: MAS-Verify**

18 $y^* \leftarrow \mathcal{A}.\texttt{Self\_Verify}(\mathcal{H})$ ;                          `// Select final answer from all candidates`
19 **return** $y^*$;

---

## B  IMPLEMENTATIONS, BENCHMARKS AND BASELINES DETAILS

**Implementation details.** As described in Sec. 4, MAS-ZERO produces 9 candidate answers. For fair comparison, we sample 9 independent outputs for CoT-SC and take the majority vote. Similarly, both debate and self-refine are run for 9 rounds. All models are accessed through their respective APIs.[8] Temperature for meta-agent is set to 0.5. For baselines, we strictly use parameters found in original papers and provided code.

**Benchmarks** Table B.1 shows the detailed statistics for each dataset. For BrowseComp and Frames, we randomly sample 10% for testing, due to the large dataset size. We evaluate SWE using its official code available at `https://github.com/SWE-bench/SWE-bench/`.

| Split | AIME24 | GPQA | SWE | BrowseComp | Frames |
|---|---|---|---|---|---|
| **Validation** | 6 | 32 | 60 | — | — |
| **Test** | 24 | 166 | 240 | 126 | 82 |

Table B.1: Data size for each split in each dataset.

**Baselines details** We use the official implementations of all baselines, sourced directly from their public repositories. For manual MAS methods, this includes ReConcile (`https://github.com/dinobby/ReConcile`), MAD (`https://github.com/Skytliang/Multi-Agents-Debate`), SPP (`https://github.com/MikeWangWZHL/Solo-Performance-Prompting`), and DyLAN (`https://github.com/SALT-NLP/DyLAN`). For automatic MAS methods, this includes ADAS (`https://github.com/ShengranHu/ADAS`), AFlow (`https://github.com/FoundationAgents/MetaGPT/tree/main/examples/aflow`), and MaAS (`https://github.com/bingreeky/MaAS`).

---

[8]We use TogetherAI API (`https://www.together.ai/`) for Llama and Qwen.

## C  COST COMPUTATION

In Fig. 1, we report the cost of single-agent systems, manual MAS, automatic MAS, and MAS-ZERO. Costs are computed using OpenAI's official pricing as of May 2025 at `https://openai.com/api/pricing/`. To ensure accuracy, we track usage directly via the official OpenAI API field "`response.usage`" for all methods. As a result, the reported values reflect the actual monetary cost, accounting for both input and output tokens.

## D  ADDITIONAL RELATED WORK

In Sec. 2, we briefly introduced the most important related works and highlighted their contrast with MAS-ZERO. In this section, we provide a more detailed discussion of existing works. For completeness, we also note that a number of training-based approaches have been proposed, but we omit them from Sec. 2 since MAS-ZERO does not involve updating LLM parameters.

Some prior work treats prompt optimization for individual agents as part of MAS design. Examples include PromptBreeder (Fernando et al., 2024), DsPy (Khattab et al., 2023) and Self-Discover (Zhou et al., 2024). More recently, this idea has been extended to broader automatic MAS design, where prompt optimization is included either as an additional design step or as part of the search space.

**Manual MAS design.**  In addition to the approaches discussed in Sec. 2, several other methods fall into this family. DyLAN Liu et al. (2024b) uses message passing to dynamically activate agent compositions; Reconcile (Chen et al., 2024b) and MAD (Liang et al., 2023) employ debate and round-table discussion, SPP (Wang et al., 2023b) leverages collaboration among multiple personas.

**Automatic MAS design.**  We follow the categories introduced in Sec. 2 and additionally include the training-based family.

**Val-Pruning.** This line of work has evolved quickly (Zhang et al., 2024; 2025b; Hu et al., 2024). Earlier examples include GPTSwarm (Zhuge et al., 2024) which optimizes graph structures via reinforcement learning but struggles to represent workflows with conditional state due to limitations of static graphs. AgentSquare (Shang et al., 2024) leverages a verifier as a performance predictor to guide the pruning.

**Val-Generation.** Besides the approaches introduced in Sec. 2 that employ a meta-agent to generate building block connections and configurations, another line of work uses the meta-agent to directly generate the required agents or blocks. For example, AutoAgents (Chen et al., 2024a) and AgentVerse (Chen et al., 2023) create specialized agents through a planner agent, while EvoAgent (Yuan et al., 2024) applies evolutionary algorithms to optimize this generation process. Similarly, Symbolic-MoE (Chen et al., 2025) leverages validation signals to construct block profiles and select the best-performing planner agent.

**Training-based.** More recent work attempts to explicitly train the agents or meta-agents within MAS. For example, ReMA (Wan et al., 2025) and OWL (Hu et al., 2025a) train the agents in a manual MAS, while MAS-GPT (Ye et al., 2025) collects data from off-the-shelf MAS and trains a meta-agent via supervised fine-tuning (SFT). FlowReasoner (Gao et al., 2025a) builds on this by extending SFT with reinforcement learning (RL). Puppeteer (Dang et al., 2025) directly RL trains the meta-agent in an end-to-end manner.

## E  STANDARD DEVIATION FOR THE EXPERIMENTS

To confirm the statistical significance of the experimental results in Table 1, we repeat the experiment *three* times, following (Zhang et al., 2025c; Liu et al., 2024b). We can see that the MAS can exhibit high variance due to the inherent nature of multi-agent systems: the variance may be amplified by the interactions among multiple agents (Cemri et al., 2025), and the generated temperature of the agents is typically non-zero. Nevertheless, MAS-ZERO is significantly stronger than other baselines, with only one exception in AIME24 and two in GPQA, as mentioned in Sec. 4.

| LLMs | GPT-4o | Llama3.3 | Qwen2.5 |
|---|---|---|---|
| CoT | ±1.97 | ±1.96 | ±0.00 |
| CoT-SC | ±3.40 | ±5.20 | ±1.96 |
| Debate | ±7.08 | ±7.08 | ±5.20 |
| Self-Refine | ±3.93 | ±1.97 | ±1.97 |
| ReConcile | ±1.97 | ±1.96 | ±1.97 |
| MAD | ±1.96 | ±3.40 | ±3.40 |
| SPP | ±5.20 | ±1.96 | ±5.89 |
| DyLAN | ±1.96 | ±0.00 | ±3.93 |
| ADAS | × | ±5.30 | ±6.38 |
| AFlow | ±1.96 | ±1.96 | ±0.00 |
| MAS-GPT | ±3.54 | ±1.96 | ±5.20 |
| MAS-ZERO | ±5.89 | ±3.15 | ±5.20 |

Table E.1: Standard deviations for AIME24.

| LLMs | GPT-4o | Llama3.3 | Qwen2.5 |
|---|---|---|---|
| CoT | ±1.29 | ±0.89 | ±2.72 |
| CoT-SC | ±1.36 | ±0.78 | ±1.94 |
| Debate | ±0.41 | ±0.78 | ±1.07 |
| Self-Refine | ±2.08 | ±2.08 | ±3.01 |
| ReConcile | ±1.29 | ±2.43 | ±1.85 |
| MAD | ±1.28 | ±1.28 | ±3.08 |
| SPP | ±1.74 | ±1.25 | ±1.20 |
| DyLAN | ±3.44 | ±0.49 | ±0.75 |
| ADAS | ±3.83 | ±3.83 | ±3.98 |
| AFlow | ±1.70 | ±2.48 | ±1.77 |
| MAS-GPT | ±1.02 | ±1.02 | ±2.25 |
| MAS-ZERO | ±1.67 | ±0.51 | ±2.08 |

Table E.2: Standard deviations for GPQA.

## F   ILLUSTRATION OF MAS-EVOLVE

Fig. F.1 illustrates MAS-Evolve. Given a question and the building blocks, the meta-agent is prompted to decompose the task and propose a MAS (see Appendix H for detailed prompts). The meta-agent then generates a MAS in the form of code, which is executed by an external compiler to produce intermediate and final outputs for the sub-tasks and agents.

After this meta-design and execution, the meta-feedback phase begins. In this phase, both the resulting MAS and its intermediate outputs are provided to the meta-agent to review their *solvability* and *completeness*. Based on this evaluation, the meta-agent generates targeted feedback. The MAS, its intermediate outputs, and the feedback are stored in the *experience library*, which is then used as additional context to refine the design in subsequent iterations.

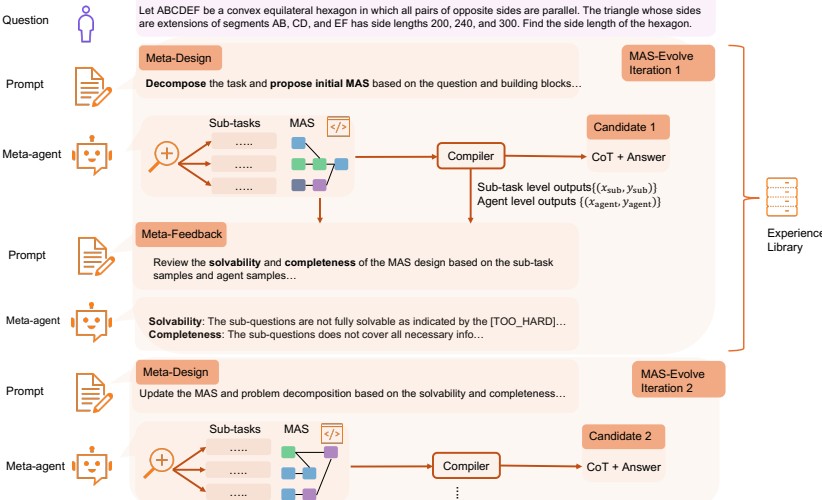

Figure F.1: Illustration for the iterations in MAS-Evolve.

## G   EXAMPLE OF MAS PRODUCED FROM MAS-ZERO

MAS-ZERO learns to decompose a new question and assign appropriate sub-MAS to each sub-task dynamically. This type of dynamic assignment would have been difficult to design manually. Fig. G.1 showcases the effectiveness of MAS-ZERO by demonstrating how it can construct and refine MAS architectures on the fly, adapting complexity to the requirements of the task.

## H   PROMPT DETAILS

As described in Sec. 3, MAS-ZERO uses prompts to implement its three steps. In this section, we present all detailed prompts used, including the building blocks (Fig. J.2, J.3, J.4, J.5) in MAS-Init;

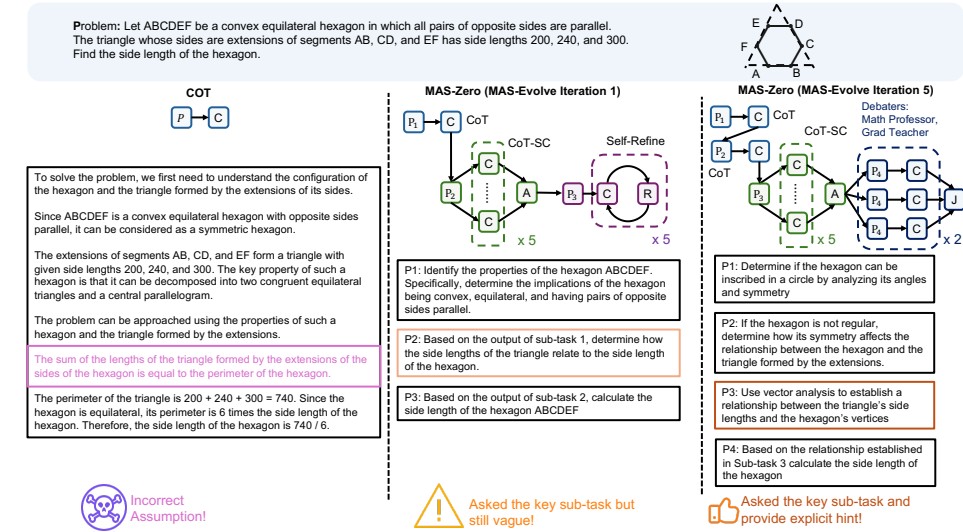

Figure G.1: An example illustrates how the MAS produced by MAS-ZERO outperforms both CoT and early iterations. In this case, MAS-ZERO learns to decompose the task into 4 sub-tasks and dynamically assign appropriate sub-MAS: CoT for the first two, CoT-SC (sampling 5 completions) for the third, and Debate (2 rounds with a math professor and a graduate teacher as debaters) for the fourth.

the Meta-Design (Fig. H.1) and Meta-Feedback (Fig. H.2) in MAS-Evolve; and the prompts for MAS-Verify (Fig. H.3).

# I    CODE TEMPLATE

MAS-ZERO uses a code template to aid MAS code generation to filling in a specific `forward` function. Fig. J.1 shows how the utility code is provided. Fig.s J.2, J.3, J.4, and J.5 show the implementations of each building blocks.

# J    USAGE OF LARGE LANGUAGE MODELS

We use LLMs solely for grammar polishing.

> ### Prompt for `Meta-Design`
>
> **Overview.** You are an expert machine learning researcher testing various agentic systems. You are given a set of building blocks (e.g., CoT, CoT-SC, Self-Refine, Debate) and a question. Each building block is represented as executable code and can contain one or more LLM agents with specialized settings (instruction, temperature, etc.). A *sub-MAS* refers to one or more building blocks assigned together to solve a specific sub-task. The overall MAS is formed by coordinating multiple sub-MAS to solve the full problem.
>
> **Your objectives are:**
>
> - **Perform task decomposition.** Decompose the given question into *manageable yet interdependent* sub-tasks, such that each sub-task is specific and detailed enough for a sub-MAS (formed from building blocks) to solve. Do not solve the tasks yourself or leak the expected answer. Instead, design the decomposition so that the sub-tasks are easier for sub-MAS to solve, and justify how they combine to yield the final answer.
>   **Make sure**
>
>   – Each sub-task should explicitly build on the outputs of prior sub-tasks.
>   – The final sub-task should naturally yield the overall answer to the original question.
>
> - **Design sub-MAS assignments.** Based on the resulting sub-tasks, assign one or more building blocks to form a sub-MAS for each sub-task. You may adjust block parameters (e.g., temperature, number of debate rounds), but you must not invent new blocks or prune the provided ones without grounding.
>   **For example:** Given available building blocks {CoT, CoT-SC, Self-Refine, Debate}, and the resulting sub-tasks: sub-task 1, sub-task 2, sub-task 3:
>
>   – **Step 1: For each sub-task, specify its sub-MAS.**
>     * It may use a **single block** (e.g., Sub-task 1: CoT).
>     * It may use a **sequential chain of blocks** (e.g., Sub-task 3: CoT → Self-Refine).
>     * Or it may use **parallel blocks inside the sub-task** (e.g., Sub-task 2: {CoT ∥ Debate}, meaning both blocks process the same input).
>
>   – **Step 2: Connect the sub-tasks (sub-MAS).** After defining sub-MAS, specify how the sub-tasks depend on one another:
>     * **Sequential connection**: Sub-tasks are connected in a linear chain, where the output of one becomes the input to the next.
>       Example: [CoT] (sub-task 1) → [Debate] (sub-task 2) → [CoT → Self-Refine] (sub-task 3).
>     * **Parallel connection**: Multiple sub-tasks depend on the same earlier sub-task's output and run in parallel.
>       Example: [CoT] (sub-task 1) → {[Debate] (sub-task 2), [CoT → Self-Refine] (sub-task 3)}. Here, both sub-task 2 and sub-task 3 consume the result of sub-task 1 in parallel.
>
>     **IMPORTANT:** Do not collapse all decomposed sub-tasks into a single instruction handled by one block. Each sub-task must be addressed by its own sub-MAS.
>
> **Final remark:** Your aim is to design an optimal block connection that can perform well on each sub-task. Your code should implement the existing blocks as-is. Do not propose new blocks or modify existing ones; you may only adjust their connections and settings (e.g., instruction, temperature).

Figure H.1: Prompt for `Meta-Design`. Additional examples, building blocks code and output format instruction are omitted for clarity.

---

**Prompt for `Meta-Feedback`**

**Overview.** You are given a candidate Multi-Agent System (MAS) design, including: (i) its executable code, (ii) the sub-task outputs from each sub-MAS, (iii) the outputs of individual agents, (iv) the final response, and (v) experience of prior iterations. Your task is to critically evaluate this MAS and provide feedback to guide refinement in the next iteration.

- **Solvability**: Assess whether all sub-tasks are solvable by the corresponding block via checking the output answer of each sub-task.
  - If the output explicitly includes [TOO_HARD], this means the sub-task is too difficult and should be further decomposed.
  - If the output is incorrect, identify whether the issue is due to
    * Insufficient decomposition (the task is still too hard) or
    * Some agents in the block are malfunctioning, or the underlying LLM is too weak to solve the sub-task. This can be checked by reviewing the agent outputs: (a) If the agent itself is not optimal (e.g., poor instruction, temperature, etc.), the settings need to be improved. (b) If the agent architecture is not optimal, a new block should be proposed by recombining existing blocks or adjusting their settings.

    Please **justify** whether it is (i), the decomposition issue or (ii) the block and agent issue. It could also be both. When proposing new sub-task, **make sure**
    * It is specific and detailed enough to solve and to contribute to the final answer;
    * All required information is carried over from previous sub-tasks or provided in the instruction;
    * The outputs from related sub-tasks are correctly incorporated (e.g., added to the `taskInfo` list when calling the agent);
    * The connection to prior sub-tasks is explicit (e.g., instructions should state "Based on the output of sub-task $i$").

- **Completeness:** Do the sub-tasks include all necessary information from the original query that can ensure the aggregation of sub-task responses can effectively yield a comprehensive answer to the user query? Note that while a sub-task might include only part of the necessary information, it is not allowable for any particular piece of critical information to be omitted from all sub-tasks.
  - If critical information is missing, refine the decomposition to include it.
  - Ensure sub-tasks are connected so that the aggregated outputs can yield a correct and comprehensive final answer.

Now, you need to improve or revise the implementation, or implement the new proposed MAS based on the reflection.

Figure H.2: Prompt for `Meta-Feedback`. Additional examples, building blocks code and output format instruction are omitted for clarity.

---

**Prompt for `MAS-Verify`**

Given the problem and a list of candidate answers, carefully review the reasoning steps and final answers to select the most reliable candidate. Do not solve the task yourself.
In your output, use the "thinking" field to compare the selected answer with each unselected answer one by one, identify the erroneous steps in the unselected answers, and give a detailed explanation of why they are incorrect. In the "selection" field, output the ID of the best answer.

Problem: {problem}
Answer List: {candidate answers}

Figure H.3: Prompt for `MAS-Verify`. The "candidate answers" have already been ranked and filtered before being passed to the meta-agent. Additional examples and output format instructions are omitted for clarity.

**Utility Code**

```python
# Named tuple for holding task information
Info = namedtuple('Info', ['name', 'author', 'content', 'prompt', 'sub_tasks', 'agents'
    , 'iteration_idx'])

# Format instructions for LLM response
FORMAT_INST = lambda request_keys: f"Reply EXACTLY with the following JSON format.\n{
    str(request_keys)}\nDO NOT MISS ANY FIELDS AND MAKE SURE THE JSON FORMAT IS
    CORRECT!\n"

# Description of the role for the LLM
ROLE_DESC = lambda role: f"You are a {role}."

class LLMAgentBase():

    def __init__(self, output_fields: list, agent_name: str,
                    role='helpful assistant', model=None, temperature=None) -> None:
        self.output_fields = output_fields
        self.agent_name = agent_name

        self.role = role
        self.model = model
        self.temperature = temperature
        # give each instance a unique id
        self.id = random_id()

    def generate_prompt(self, input_infos, instruction) -> str:
        # generate prompt based on the input_infos
        # ...

    def query(self, input_infos: list, instruction, iteration_idx=-1) -> dict:
        # call generate_prompt and the LLM to get output
        # ...

    def __repr__(self):
        return f"{self.agent_name} {self.id}"

class AgentArchitecture:
    """
    Fill in your code here.

    def forward(self, taskInfo) -> Union[Info, str]:
        Args:
        - taskInfo (Info): Task information.

        Returns:
        - Answer (Info): Your FINAL Answer.
    """
```

Figure J.1: Utility code.

**Implementation of CoT Building Blocks**

```python
def forward(self, taskInfo):
    # Instruction for the Chain-of-Thought (CoT) approach
    # It is an important practice that allows the LLM to think step by step before
        solving the task.
    cot_instruction = self.cot_instruction

    # Instantiate a new LLM agent specifically for CoT
    # To allow the LLM to think before answering, we need to set an additional output
        field 'thinking'.
    cot_agent = LLMAgentBase(['thinking', 'answer'], 'Chain-of-Thought Agent', model=
        self.node_model, temperature=0.0)

    # Prepare the inputs for the CoT agent
    # The input should be a list of Info, and the first one is often the taskInfo
    cot_agent_inputs = [taskInfo]

    # Get the response from the CoT agent
    thinking, answer = cot_agent(cot_agent_inputs, cot_instruction)
    final_answer = self.make_final_answer(thinking, answer)

    # Return only the final answer
    return final_answer
```

Figure J.2: Implementation of CoT building blocks

**Implementation of CoT-SC Building Blocks**

```python
def forward(self, taskInfo):
    # Instruction for step-by-step reasoning
    cot_instruction = self.cot_instruction
    N = self.max_sc # Number of CoT agents

    # Initialize multiple CoT agents with a higher temperature for varied reasoning
    cot_agents = [LLMAgentBase(['thinking', 'answer'], 'Chain-of-Thought Agent', model
        =self.node_model, temperature=0.5) for _ in range(N)]

    # Majority voting function to select the most common answer
    from collections import Counter
    def majority_voting(answers):
        return Counter(answers).most_common(1)[0][0]

    thinking_mapping = {}
    answer_mapping = {}
    possible_answers = []
    for i in range(N):
        thinking, answer = cot_agents[i]([taskInfo], cot_instruction)
        possible_answers.append(answer.content)
        thinking_mapping[answer.content] = thinking
        answer_mapping[answer.content] = answer

    # Ensembling the answers from multiple CoT agents
    answer = majority_voting(possible_answers)

    thinking = thinking_mapping[answer]
    answer = answer_mapping[answer]

    final_answer = self.make_final_answer(thinking, answer)

    return final_answer
```

Figure J.3: Implementation of CoT-SC building blocks

**Implementation of Debate Building Blocks**

```python
def forward(self, taskInfo):
    # Instruction for initial reasoning
    debate_initial_instruction = self.cot_instruction

    # Instruction for debating and updating the solution based on other agents'
        solutions
    debate_instruction = "Given solutions to the problem from other agents, consider
        their opinions as additional advice. Please think carefully and provide an
        updated answer. Put your thinking process in the 'thinking' field and the
        updated answer in the 'answer' field. "

    # Initialize debate agents with different roles and a moderate temperature for
        varied reasoning
    debate_agents = [LLMAgentBase(['thinking', 'answer'], 'Debate Agent',  model=self.
        node_model, role=role, temperature=0.5) for role in self.debate_role]

    # Instruction for final decision-making based on all debates and solutions
    final_decision_instruction = "Given all the above thinking and answers, reason over
         them carefully and provide a final answer. Put your thinking process in the '
        thinking' field and the final answer in the 'answer' field."
    final_decision_agent = LLMAgentBase(['thinking', 'answer'], 'Final Decision Agent',
         model=self.node_model, temperature=0.0)

    max_round = self.max_round # Maximum number of debate rounds
    all_thinking = [[] for _ in range(max_round)]
    all_answer = [[] for _ in range(max_round)]

    # Perform debate rounds
    for r in range(max_round):
        for i in range(len(debate_agents)):
            if r == 0:
                thinking, answer = debate_agents[i]([taskInfo],
                    debate_initial_instruction)
            else:
                input_infos = [taskInfo] + [all_thinking[r-1][i]] + all_thinking[r-1][:
                    i] + all_thinking[r-1][i+1:]
                thinking, answer = debate_agents[i](input_infos, debate_instruction)
            all_thinking[r].append(thinking)
            all_answer[r].append(answer)

    # Make the final decision based on all debate results and solutions
    thinking, answer = final_decision_agent([taskInfo] + all_thinking[max_round-1] +
        all_answer[max_round-1], final_decision_instruction)
    final_answer = self.make_final_answer(thinking, answer)

    return final_answer
```

Figure J.4: Implementation of Debate building blocks

**Implementation of Self-Refine Building Blocks**

```python
def forward(self, taskInfo):
    # Instruction for initial reasoning
    cot_initial_instruction = self.cot_instruction

    # Instruction for reflecting on previous attempts and feedback to improve
    cot_reflect_instruction = "Given previous attempts and feedback, carefully consider
        where you could go wrong in your latest attempt. Using insights from previous
        attempts, try to solve the task better."
    cot_agent = LLMAgentBase(['thinking', 'answer'], 'Chain-of-Thought Agent', model=
        self.node_model, temperature=0.0)

    # Instruction for providing feedback and correcting the answer
    critic_instruction = "Please review the answer above and criticize on where might
        be wrong. If you are absolutely sure it is correct, output exactly 'True' in '
        correct'."
    critic_agent = LLMAgentBase(['feedback', 'correct'], 'Critic Agent', model=self.
        node_model, temperature=0.0)

    N_max = self.max_round # Maximum number of attempts

    # Initial attempt
    cot_inputs = [taskInfo]
    thinking, answer = cot_agent(cot_inputs, cot_initial_instruction, 0)

    for i in range(N_max):
        # Get feedback and correct status from the critic
        feedback, correct = critic_agent([taskInfo, thinking, answer],
            critic_instruction, i)
        if correct.content == 'True':
            break

        # Add feedback to the inputs for the next iteration
        cot_inputs.extend([thinking, answer, feedback])

        # Reflect on previous attempts and refine the answer
        thinking, answer = cot_agent(cot_inputs, cot_reflect_instruction, i + 1)

    final_answer = self.make_final_answer(thinking, answer)

    return final_answer
```

Figure J.5: Implementation of Self-Refine building blocks

