# OpenReview forum: "MAS-Zero: Designing Multi-Agent Systems with Zero Supervision"
_ICLR.cc/2026/Conference — ICLR 2026 Conference Withdrawn Submission_

### Official Review · Reviewer_mcvj · 2025-10-15

**Soundness:** 2
**Presentation:** 3
**Contribution:** 2
**Rating:** 2
**Confidence:** 4

**Summary:**

This paper introduces MAS-ZERO, a novel framework for automatically designing Multi-Agent Systems at inference time without requiring a supervised validation set. The core of the system is a meta-agent that iteratively refines an MAS configuration for each specific problem instance. The process involves three stages: MAS-Init, which generates baseline solutions from a set of predefined "building block" strategies; MAS-Evolve, where the meta-agent decomposes the problem, designs an MAS, and refines it based on self-generated feedback on sub-task solvability and completeness; and MAS-Verify, which selects the best final answer from all generated candidates. The authors demonstrate that MAS-ZERO achieves state-of-the-art performance across challenging reasoning, coding, and agentic benchmarks, establishing a new Pareto frontier for the accuracy vs. cost trade-off.

**Strengths:**

1. The motivations are very great. For example, this paper wants to eliminate the need for a labeled validation set.
2. The framework's explicit mechanism for breaking down complex problems into manageable sub-tasks and designing tailored sub-MAS for each is a robust approach to tackling intricate challenges.
3. The paper can be easily understood.

**Weaknesses:**

1. The framework centralizes a heavy cognitive load on the meta-agent, which must simultaneously evaluate agent capabilities, refine system architecture, and verify final answers. This creates a potential single point of failure and places a high floor on the required capability of the underlying LLM.

2. While the framework avoids a traditional validation set, it replaces it with an online, self-referential validation loop (Meta-Feedback and MAS-Verify). This raises concerns about evaluation blind spots or self-reinforcing biases, as the "judge" is the same model responsible for the design.

3. The system's creative scope is limited by the initial, manually-provided set of building blocks. The meta-agent primarily optimizes the composition and parameters of these existing modules rather than inventing fundamentally new agentic behaviors or interaction patterns.

4. The iterative, multi-candidate generation process incurs substantial computational cost at inference time for every single problem. While presented as a favorable trade-off, this per-instance expense could be prohibitive for many practical, large-scale applications.

**Questions:**

1. How does the performance gap between MAS-ZERO and simpler baselines (e.g., CoT-SC) evolve as the underlying LLM's capability increases? Does the sophisticated design process yield diminishing or increasing returns with more powerful models?

2. Did the self-evolution process uncover any surprising or counter-intuitive MAS designs that challenge conventional human intuition? Are there examples of novel agent compositions or problem decompositions that were consistently discovered for specific task archetypes?

3. As single models become more capable and integrate complex reasoning and planning internally, how do you see the role of external multi-agent orchestration frameworks evolving? Will their functions be absorbed into the base models, or will systems like MAS-ZERO remain essential?

---

> ### Author Response · Authors · 2025-11-18
> **Response to Reviewer mcvj (Part 1)**
>
> Thank you very much for recognizing the motivation, robustness, and clarity of our work. We also appreciate your constructive feedback. Please find our point-to-point responses below.
>
> > W1: The framework centralizes a heavy cognitive load on the meta-agent, which must simultaneously evaluate agent capabilities, refine system architecture, and verify final answers. This creates a potential single point of failure and places a high floor on the required capability of the underlying LLM.
>
> This is a key trade-off of any self-evolving system when compared to validation-based and training-based methods. We believe that LLMs will continue to improve in instruction following, planning, critiquing, and judging. For example, post-training for instruction following is now a basic requirement for modern LLMs. LLM-as-a-judge is an active research area and is already common a standard for assessing foundation models. Therefore, we think that the abilities needed for an LLM to serve as a meta-agent are already present in current models and will become increasingly accessible.
>
> We show the overall improving performance in Table 1 as evidence that this is achievable even with today’s LLM capability. To make this clearer, we manually examine the 24 test cases in AIME24 (all trajectories will be publicly released) using GPT-4o as the meta-agent. We compare each case with the ground-truth solution (a common method to check alignment, e.g., [2]) and label the decomposition and critique quality for each test instance as one of {Strong, Moderate, Weak}. Specifically:
>
> (1) **Decomposition**: A strong decomposition mirrors the true solution or reflects clear dependencies between sub-tasks. A weak decomposition is broad or vague (for example, “solve the sub-task via parameterization”), with unclear dependencies or simple rephrasing of an existing sub-task.
>
> (2) **Critique**: A strong critique identifies which sub-task fails, traces dependency chains, identifies architectural mistakes (for example, noting inconsistencies across CoT-SC and inferring that the sub-task is too hard or unreliable), and provides actionable feedback. A weak critique is generic or consists only of short acknowledgments without new insight.
>
> | Meta-agent's tasks | Strong | Moderate | Weak |
> | -------- | -------- | -------- | -------- |
> | Decomposition     | 9     | 12     | 3 |
> | Critique     | 22     | 2     | 0 |
>
>
> The table above shows the number of examples for each label based on our manual inspection. We see that the current LLM (GPT-4o) already matches human expectation in critique and reflection.  We believe these numbers will be even higher with newest models like GPT 5 or Gemini 3. This is also consistent with prior work, where LLMs are often used to judge or critique (e.g, [1] uses an LLM to judge question difficulty). We also note that decomposition remains more challenging. With increasing LLM capability, we expect further improvements in both decomposition and critique.
>
> > W2: While the framework avoids a traditional validation set, it replaces it with an online, self-referential validation loop (Meta-Feedback and MAS-Verify). This raises concerns about evaluation blind spots or self-reinforcing biases, as the "judge" is the same model responsible for the design.
>
> We agree that this is an inherent challenge in any system that relies on LLM-based judging. We see this as an advantage rather than a limitation. MAS-Zero provides a direct way to use stronger judge models when they become available in the future, which is a useful and future-proof property. As mentioned in L290–L297, one key challenge in verification is position bias, a well-known issue in the LLM-as-judge community [1, 2]. In MAS-Zero, we adopt a prior that favors majority responses. This simple approach helps reduce the bias, although the LLM still shows noticeable position bias. We also show the performance gap that could be closed if a stronger verifier were available, indicating clear room for improvement once better judge models are accessible.
>
> [1]: A survey on llm-as-a-judge, Gu et al., 2025.
> [2]: Evaluating judges as evaluators: The jetts benchmark of llm-as-judges as test-time scaling evaluators, Zhou et al., 2025.
>
> > W3: The system's creative scope is limited by the initial, manually-provided set of building blocks. The meta-agent primarily optimizes the composition and parameters of these existing modules rather than inventing fundamentally new agentic behaviors or interaction patterns.
>
> This is intentional. Our goal in MAS-Zero is not to propose new agent behaviors or new interaction patterns, but to find a good composition for a given question. In L227–L231, we explain that creating new agent behaviors would lead to a prohibitively large action space, and our pilot study also shows sub-optimal results in this setting. We believe that a MAS design grounded in the provided building blocks is a more suitable choice.

---

> ### Author Response · Authors · 2025-11-18
> **Response to Reviewer mcvj (Part 2)**
>
> > W4: The iterative, multi-candidate generation process incurs substantial computational cost at inference time for every single problem. While presented as a favorable trade-off, this per-instance expense could be prohibitive for many practical, large-scale applications.
>
> Fig 1 shows the trade-off between effectiveness and efficiency "includes both 'training' (if any) and test-time usage." (L378-L385). We can see that MAS-Zero delivers high performance at lower cost than the considered baselines, establishing a new frontier for this trade-off. We believe this is a fair comparison because the central idea of any inference-time method (including MAS-Zero) is to shift part of the expensive optimization cost into the inference phase (L128–L130). It is expected that such methods will cost more than using only the inference phase of an optimization-based method, as the full cost of an optimization-based method includes both optimization and inference. Additionally, optimization-based methods require in-domain validation sets or training sets, which are often unavailable and may not generalize
>
> In the scenario where a user cares strongly about reducing test-time cost, MAS-Zero can disable MAS-Evolve. In this case, instead of running "4 INIT + 5 EVOLVE + 1 JUDGE", it only needs "4 INIT + 1 JUDGE", which is much cheaper. The table below shows the test-time cost of the optimization-based baselines and MAS-Zero, using GPT-4o as the underlying LLM. We see that while MAS-Zero without MAS-Evolve (MAS-Zero (–MAS-Evolve)) has lower performance than full MAS-Zero, it still outperforms (or matches) the other baselines while using far less cost (smaller than the considered baselines). This demonstrates that MAS-Zero is cost-efficient and can adapt to different scenarios.
>
>
> | Method | Inference-time Cost ($) | AIME24 | GPQA |
> | -------- | -------- | -------- | -------- |
> | MaAS     | 18.34     | 12.50     | 43.37 |
> | ADAS     | 6.19     | 0.0     | 45.20 |
> | AFlow     | 5.46     | 20.83     | 46.99 |
> | MAS-Zero     | 15.68     | 33.33     | 50.63 |
> | MAS-Zero (-MAS-Evolve)  | 4.03   | 20.00     | 48.73     |
>
> > Q1: How does the performance gap between MAS-ZERO and simpler baselines (e.g., CoT-SC) evolve as the underlying LLM's capability increases? Does the sophisticated design process yield diminishing or increasing returns with more powerful models?
>
> Table 4 is designed for this. We can see that MAS-Zero still improves in stronger underlying LLMs (o3-mini). Comparing to weaker LLM (GPT-4o) in Table 1, the gap is smaller (10%+ gap in GPT-4o while ~10% in o3-mini). On average, MAS-Zero still outperforming baselines and generalize well to stronger LLM.
>
> > Q2: Did the self-evolution process uncover any surprising or counter-intuitive MAS designs that challenge conventional human intuition? Are there examples of novel agent compositions or problem decompositions that were consistently discovered for specific task archetypes?
>
> Yes. There are two key surprising and interesting findings in architecture and decomposition:
>
> (1) **Single-agent and simple MAS (the 4 building blocks) can outperform sophisicated MAS in some scenarios** (Fig 1). This is surprising, as one might assume that MAS, with complex composition designs, should always perform better. We further confirm this in the Table 6 ablation, which shows that the final verification stage is the most significant among the three stages, giving MAS-Zero the flexibility to reduce to the building blocks. This urges future researchers to more systematically consider when MAS should be used and when it should not.
>
>
> (2) **Sub-task decomposition behaviors lies on a spectrum**: We observe a wide spectrum of decomposition behaviors (all evolving trajectories will be publicly available). In an extreme case, the meta-agent solves the task itself or delegates the entire task to a single sub-agent (for example, a sub-task explicitly instructed to “decompose the task”). In the middle range, where the meta-agent decomposes the task in a more standard way, we often find that one sub-task carries most of the workload. This is surprising, as it reveals potential hidden biases in task decomposition that we did not anticipate. These findings could serve as evidence to guide future improvements.

---

> ### Author Response · Authors · 2025-11-18
> **Response to Reviewer mcvj (Part 3)**
>
> > Q3: As single models become more capable and integrate complex reasoning and planning internally, how do you see the role of external multi-agent orchestration frameworks evolving? Will their functions be absorbed into the base models, or will systems like MAS-ZERO remain essential?
>
>
> Thank you for raising this insightful question! It is an important topic, and the MAS community has not yet reached a clear answer. MAS-Zero shows that a single agent can be surprisingly strong in some scenarios, and MAS is not always preferable. We view MAS-Zero as a pilot study that explores this question. We believe the answer is not a simple yes/no, but instead depends on several factors that may affect the effectiveness of MAS (including MAS-Zero). For example:
>
> (1) **Reasoning structure of the question**: Some questions have underlying dependency structures, such as a dependency graph. A single agent *may or may not* capture these structures internally, depending on the model capability. In contrast, MAS can be helpful because it can explicitly provide the structure in the proposed MAS.
>
> (2) **Distinction of the underlying sub-tasks**: When the sub-tasks are highly coupled, a single agent may be a better choice. When the sub-tasks are distinct or even conflicting (e.g., different search queries involved), MAS can handle them separately and is often preferable.
>
> (3) **Underlying LLM capacity**. When the underlying LLM is strong, many functions of MAS may already be internalized, such as the reasoning structure, and a single agent may be sufficient. However, strong LLMs usually consume far more tokens, and one key advantage of MAS is that it can make use of multiple context windows through multiple sub-agents, serving as a form of context management. In that case, MAS can be preferable.
>
> We urge the community to study this question in a more systematic way, and we hope MAS-Zero can serve as a good starting point for this line of exploration.

---

### Official Review · Reviewer_R95u · 2025-11-01

**Soundness:** 3
**Presentation:** 3
**Contribution:** 3
**Rating:** 4
**Confidence:** 4

**Summary:**

This work proposes MAS-ZERO, a self-evolved, inference-time framework that requires no validation set and can iteratively optimize Multi-Agent System (MAS) configurations through meta-level design, which enabling dynamic task decomposition and agent composition, simplifies the system when appropriate, and outperforms both manual and existing automatic MAS baselines across multiple tasks and different Large Language Models (LLMs).

**Strengths:**

1. Originality: a new zero-supervision, inference-time self-evolving MAS framework, breaking validation set dependence and supporting dynamic task decomposition/complexity switching.
2. Quality: Comprehensive experiments across domains/models, comparisons with 11 baselines, ablation studies validating core modules, and planned open-sourcing ensuring reproducibility.
3. Clarity: Clearly describes the framework (3 key steps) and details (code templates, prompts), with complete appendices reducing understanding barriers.
4. Significance: Outperforms baselines on high-difficulty tasks (AIME24, SWE) and reveals the new insight that "simpler systems excel in some scenarios," with practical and guiding value.

**Weaknesses:**

1. From my understanding, this paper focuses on prompt engineering and involves no model training. Its performance upper bound is constrained by the capabilities of the underlying model, and its effectiveness bears similarities to test-time scaling. I consider its contributions limited, that is, had the authors proposed a training paradigm or a data framework to explore ways of pushing the model’s performance ceiling, the work would have been far more impactful.
2. Drawing parallels to test-time scaling, increasing the inference budget will undoubtedly yield performance gains. I believe presenting a table that quantifies the trade-off between effectiveness and efficiency (e.g., latency, token consumption) would be highly valuable. It appears that time consumption surges with the system’s complexity, which poses a relatively significant challenge for user experience and real-world robotics deployment.
3. The experiments lack in-depth analysis. For instance, there is no exploration of the specific patterns or scenarios where different methods fail.

**Questions:**

Address my weeknesses.

---

> ### Author Response · Authors · 2025-11-18
> **Response to Reviewer R95u (Part 1)**
>
> Thank you very much for recognizing the novelty, comprehensiveness, and clarity of our work, as well as our finding regarding the effectiveness of simpler systems relative to MAS in certain scenarios. We also appreciate your constructive feedback. Please find our point-to-point responses below:
>
> > W1: From my understanding, this paper focuses on prompt engineering and involves no model training. Its performance upper bound is constrained by the capabilities of the underlying model, and its effectiveness bears similarities to test-time scaling. I consider its contributions limited, that is, had the authors proposed a training paradigm or a data framework to explore ways of pushing the model’s performance ceiling, the work would have been far more impactful.
>
> We believe both training and inference-time methods are important, and they are not contradictory. Inference-time methods can further improve a trained model, and we would like to point out that MAS-Zero outperforms common test-time scaling methods such as Debate and Self-Refinement. We also include a recent strong public training-based baseline (MAS-GPT), and MAS-Zero outperforms it by a large margin (more than 10% improvement), again showing strong effectiveness.
>
> We also believe that the scope of novelty is broad and not limited to introducing a new paradigm. It can also include extending existing paradigms to new problems, designing new experiments and ablations, and identifying new observations and insights,. In MAS-Zero, we extend inference-time methods to multi-agent system design for the first time, achieve SoTA performance, and conduct comprehensive experiments and ablations to understand MAS better (for example, single-agent and simple MAS can perform surprisingly well), showing the potential room for improvement in current designs. We believe these points all represent meaningful contributions. Prompt engineering is widely accepted and justified in the inference-time community ([1,2]), and we believe this does not reduce our contributions.
>
> [1]: SELF-DISCOVER: Large Language Models Self-Compose Reasoning Structures, Zhou et al., NeurIPS 2024.
> [2]: Gepa: Reflective prompt evolution can outperform reinforcement learning, Agrawal et al., 2025.
>
> > W2: Drawing parallels to test-time scaling, increasing the inference budget will undoubtedly yield performance gains. I believe presenting a table that quantifies the trade-off between effectiveness and efficiency (e.g., latency, token consumption) would be highly valuable. It appears that time consumption surges with the system’s complexity, which poses a relatively significant challenge for user experience and real-world robotics deployment.
>
> Fig. 1 shows the trade-off between effectiveness and efficiency, including "both 'training' (if any) and test-time usage." (L378-L385). We can see that MAS-Zero delivers high performance at lower cost than the considered baselines, establishing a new frontier for this trade-off. We believe this is a fair comparison because the central idea of any inference-time method (including MAS-Zero) is to shift the expensive optimization cost into the inference phase (L128–L130). It is expected that such methods will cost more than using only the inference phase of an optimization-based method, as the full cost of an optimization-based method includes both optimization and inference. Additionally, optimization-based methods require in-domain validation sets or training sets, which are often unavailable and may not generalize
>
> In the scenario where a user cares strongly about reducing test-time cost, MAS-Zero can disable MAS-Evolve. In this case, instead of running "4 INIT + 5 EVOLVE + 1 JUDGE", it only needs "4 INIT + 1 JUDGE", which is much cheaper. The table below shows the test-time cost of the optimization-based baselines and MAS-Zero, using GPT-4o as the underlying LLM. We see that while MAS-Zero without MAS-Evolve (MAS-Zero (–MAS-Evolve)) has lower performance than full MAS-Zero, it still outperforms (or matches) the other baselines while using far less cost (smaller than the considered baselines). This demonstrates that MAS-Zero is cost-efficient and can adapt to different scenarios.
>
> | Method | Inference-time Cost ($) | AIME24 | GPQA |
> | -------- | -------- | -------- | -------- |
> | MaAS     | 18.34     | 12.50     | 43.37 |
> | ADAS     | 6.19     | 0.0     | 45.20 |
> | AFlow     | 5.46     | 20.83     | 46.99 |
> | MAS-Zero     | 15.68     | 33.33     | 50.63 |
> | MAS-Zero (-MAS-Evolve)  | 4.03   | 20.00     | 48.73     |

---

> ### Author Response · Authors · 2025-11-18
> **Response to Reviewer R95u (Part 2)**
>
> > W3: The experiments lack in-depth analysis. For instance, there is no exploration of the specific patterns or scenarios where different methods fail.
>
>
> We provided comprehensive ablations and in-depth analysis in Sec. 4.1, showing how each step in MAS-Zero behaves when different underlying LLMs are used. We also conduct ablations on each step, demonstrating the importance, gain, and potential of each step. We would like to highlight two key and somewhat surprising findings in architecture and decomposition that reveal scenarios where MAS may fail:
>
> (1) **Single-agent and simple MAS (the 4 building blocks) can outperform sophisticated MAS in some scenarios** (Fig 1). This is surprising, as one might assume that MAS, with complex composition designs, should always perform better. We further confirm this in the Table 6 ablation, which shows that the final verification stage is the most significant among the three stages, giving MAS-Zero the flexibility to reduce to the building blocks. This urges future researchers to more systematically consider when MAS should be used and when it should not.
>
> (2) **Sub-task decomposition behavior lies on a spectrum**: We observe a wide spectrum of decomposition behaviors (all evolving trajectories will be publicly available). In an extreme case, the meta-agent solves the task itself or delegates the entire task to a single sub-agent (for example, a sub-task explicitly instructed to “decompose the task”). In the middle range, where the meta-agent decomposes the task in a more standard way, we often find that one sub-task carries most of the workload. This is surprising, as it reveals potential hidden biases in task decomposition that we did not anticipate. These findings could serve as evidence to guide future improvements.
>
> If there is any additional analysis you would like us to add, please let us know. We would be happy to include it.

---

### Official Review · Reviewer_gDSq · 2025-11-01

**Soundness:** 2
**Presentation:** 2
**Contribution:** 2
**Rating:** 4
**Confidence:** 3

**Summary:**

The authors propose MAS-ZERO, an inference-time-only framework for automatically designing Multi-Agent Systems (MAS) without supervision or validation sets. The core idea is to use a "meta-agent" that operates in a three-stage process: 1) MAS-Init, which runs a set of predefined "building block" strategies (like CoT, Debate) to gather initial solutions; 2) MAS-Evolve, an iterative loop where the meta-agent decomposes the problem, designs a novel MAS by composing the building blocks, executes it, and then critiques its own design based on "solvability" and "completeness" feedback; 3) MAS-Verify, which selects the best final answer from the pool of candidates generated by both MAS-Init and MAS-Evolve. The authors demonstrate that this self-evolving, per-instance approach outperforms both manual and prior automatic MAS baselines on tasks in reasoning, coding, and agentic benchmarks.

**Strengths:**

1. The framework's primary strength is its ability to adapt its design to each specific problem instance at inference time, without the need for a pre-tuned configuration or a validation set. This is a compelling alternative to static, validation-set-tuned systems.

2. The expanded ablation studies in Section 4.2 provide a clear and valuable breakdown of the system's performance. The ablations on MAS-Init, MAS-Evolve, meta-design, and meta-feedback effectively demonstrate that all components contribute meaningfully to the final result.

3. The method shows consistent and sometimes substantial improvements over strong baselines across multiple challenging domains (AIME24, SWE-Bench), demonstrating the practical effectiveness of the self-evolving design.

**Weaknesses:**

1. While the empirical results are now much stronger, the core conceptual framework can be viewed as a very sophisticated and well-executed combination of existing ideas (problem decomposition, self-refinement loops, agent routing) orchestrated via prompt engineering. The "meta-design" component, while effective, is fundamentally a well-crafted heuristic rather than a wholly new paradigm.

2. The entire system's success hinges on the reasoning capability of the meta-agent to correctly decompose problems and accurately critique its own designs. The paper acknowledges this the ablations and show the components are necessary, but the system is still vulnerable to meta-agent flaws. A failure in meta-feedback could send the iterative process in a useless direction.

3. For Table 5, the current experiment only swaps the meta-agent or the agents. A more powerful extension, which the framework seems suited for, would be for the meta-agent to dynamically assign sub-tasks to different agent models based on their known strengths (e.g., assign a reasoning-focused sub-task to o3-mini and a coding sub-task to a coder model), rather than using a homogeneous pool of agents.

**Questions:**

1. The cost-efficiency analysis (Fig. 1) is useful. However, for a practical application, what is the inference-time latency comparison? MAS-ZERO runs 9 total candidate generations (4 init + 5 evolve). How does this end-to-end time/cost compare to a single inference pass from a baseline like AFlow or MAS-GPT on the same test problem?

2. Figure 4B shows a massive performance leap with an oracle verifier, implying the MAS-Verify step is a significant bottleneck. Since MAS-Verify is also an LLM call, have the authors experimented with using a stronger, dedicated model (e.g., O3) only for the final verification step, even when the agents and meta-agent are weaker (e.g., GPT-4o)?

3. The system is initialized with four building blocks (CoT, CoT-SC, Debate, Self-Refine). How sensitive is the final performance to this specific set? For instance, what happens if it is only given CoT and Debate? Does the "MAS-Evolve" step successfully reinvent a Self-Refine-like process, or is its creativity constrained to what it is initially given?

---

> ### Author Response · Authors · 2025-11-18
> **Response to Reviewer gDSq (Part 1)**
>
> Thank you very much for recognizing the strength of our inference-time method, as well as our comprehensive analysis and strong empirical results. We also appreciate your constructive feedback. Please find our point-to-point responses below:
>
> > W1: While the empirical results are now much stronger, the core conceptual framework can be viewed as a very sophisticated and well-executed combination of existing ideas (problem decomposition, self-refinement loops, agent routing) orchestrated via prompt engineering. The "meta-design" component, while effective, is fundamentally a well-crafted heuristic rather than a wholly new paradigm.
>
> We believe that the scope of novelty is broad and not limited to introducing a new paradigm. It can also include extending existing paradigms to new problems, designing new experiments and ablations, and identifying new observations and insights. In MAS-Zero, we extend inference-time methods to multi-agent system design for the first time, achieve SoTA performance, and conduct comprehensive experiments and ablations to understand MAS better (for example, single-agent and simple MAS can perform surprisingly well), showing the potential room for improvement in current designs. We believe these points all represent meaningful contributions. Prompt engineering is widely accepted and justified in the inference-time community ([1,2]), and we believe this does not reduce our contributions.
>
> [1]: SELF-DISCOVER: Large Language Models Self-Compose Reasoning Structures, Zhou et al., NeurIPS 2024.
> [2]: Gepa: Reflective prompt evolution can outperform reinforcement learning, Agrawal et al., 2025
>
> > W2: The entire system's success hinges on the reasoning capability of the meta-agent to correctly decompose problems and accurately critique its own designs. The paper acknowledges this the ablations and show the components are necessary, but the system is still vulnerable to meta-agent flaws. A failure in meta-feedback could send the iterative process in a useless direction.
>
> This is a key trade-off of any self-evolving system when compared to validation-based and training-based methods. We believe that LLMs will continue to improve in instruction following, planning, critiquing, and judging. For example, post-training for instruction following is now a basic requirement for modern LLMs. LLM-as-a-judge is an active research area and is already common a standard for assessing foundation models. Therefore, we think that the abilities needed for an LLM to serve as a meta-agent are already present in current models and will become increasingly accessible.
>
> We show the overall improving performance in Table 1 as evidence that this is achievable even with today’s LLM capability. To make this clearer, we manually examine the 24 test cases in AIME24 (all trajectories will be publicly released) using GPT-4o as the meta-agent. We compare each case with the ground-truth solution (a common method to check alignment, e.g., [2]) and label the decomposition and critique quality for each test instance as one of {Strong, Moderate, Weak}. Specifically:
>
> (1) **Decomposition**: A strong decomposition mirrors the true solution or reflects clear dependencies between sub-tasks. A weak decomposition is broad or vague (for example, “solve the sub-task via parameterization”), with unclear dependencies or simple rephrasing of an existing sub-task.
>
> (2) **Critique**: A strong critique identifies which sub-task fails, traces dependency chains, identifies architectural mistakes (for example, noting inconsistencies across CoT-SC and inferring that the sub-task is too hard or unreliable), and provides actionable feedback. A weak critique is generic or consists only of short acknowledgments without new insight.
>
> | Meta-agent's tasks | Strong | Moderate | Weak |
> | -------- | -------- | -------- | -------- |
> | Decomposition     | 9     | 12     | 3 |
> | Critique     | 22     | 2     | 0 |
>
>
> The table above shows the number of examples for each label based on our manual inspection. We see that the current LLM (GPT-4o) already matches human expectation in critique and reflection. We believe these numbers will be even higher with newest models like GPT 5 or Gemini 3. This is also consistent with prior work, where LLMs are often used to judge or critique (e.g, [1] uses an LLM to judge question difficulty). We also note that decomposition remains more challenging. With increasing LLM capability, we expect further improvements in both decomposition and critique.
>
> [1]: Scaling LLM Test-Time Compute Optimally can be More Effective than Scaling Model Parameters. Snell et al., 2024.
> [2]: Reliable Fine-Grained Evaluation of Natural Language Math Proofs. Ma et al., 2025

---

> ### Author Response · Authors · 2025-11-18
> **Response to Reviewer gDSq (Part 2)**
>
> > W3: For Table 5, the current experiment only swaps the meta-agent or the agents. A more powerful extension, which the framework seems suited for, would be for the meta-agent to dynamically assign sub-tasks to different agent models based on their known strengths (e.g., assign a reasoning-focused sub-task to o3-mini and a coding sub-task to a coder model), rather than using a homogeneous pool of agents.
>
> Thank you for the important suggestion. We believe MAS-Zero opens up many interesting directions, including LLM routing (as you suggested), meta-agent training, understanding when MAS should be used, and more. We are currently working on several of these directions, including LLM routing. That being said, we believe MAS-Zero is already a solid foundational work that can, on its own, inspire many follow-up ideas.
>
> > Q1: The cost-efficiency analysis (Fig. 1) is useful. However, for a practical application, what is the inference-time latency comparison? MAS-ZERO runs 9 total candidate generations (4 init + 5 evolve). How does this end-to-end time/cost compare to a single inference pass from a baseline like AFlow or MAS-GPT on the same test problem?
>
> Fig 1 "includes both 'training' (if any) and test-time usage." (L378-L385). We believe this is a fair comparison because the central idea of any inference-time method (including MAS-Zero) is to shift the expensive optimization cost into the inference phase (L128–L130). It is expected that such methods will cost more than using **only** the inference phase of an optimization-based method, as the full cost of an optimization-based method includes both optimization and inference. Additionally, optimization-based methods require in-domain validation sets or training sets, which are often unavailable and may not generalize.
>
> In the scenario where a user cares strongly about reducing test-time cost, MAS-Zero can disable MAS-Evolve. In this case, instead of running "4 INIT + 5 EVOLVE + 1 JUDGE", it only needs "4 INIT + 1 JUDGE", which is much cheaper. The table below shows the test-time cost of the optimization-based baselines and MAS-Zero, using GPT-4o as the underlying LLM. We see that while MAS-Zero without MAS-Evolve (MAS-Zero (–MAS-Evolve)) has lower performance than full MAS-Zero, it still outperforms (or matches) the other baselines while using far less cost (smaller than the considered baselines). This demonstrates that MAS-Zero is cost-efficient and can adapt to different scenarios.
>
> | Method | Inference-time Cost ($) | AIME24 | GPQA |
> | -------- | -------- | -------- | -------- |
> | MaAS     | 18.34     | 12.50     | 43.37 |
> | ADAS     | 6.19     | 0.0     | 45.20 |
> | AFlow     | 5.46     | 20.83     | 46.99 |
> | MAS-Zero     | 15.68     | 33.33     | 50.63 |
> | MAS-Zero (-MAS-Evolve)  | 4.03   | 20.00     | 48.73     |
>
> > Q2: Figure 4B shows a massive performance leap with an oracle verifier, implying the MAS-Verify step is a significant bottleneck. Since MAS-Verify is also an LLM call, have the authors experimented with using a stronger, dedicated model (e.g., O3) only for the final verification step, even when the agents and meta-agent are weaker (e.g., GPT-4o)?
>
> Yes. This is an important finding in MAS-Zero. The verifier, which can decide whether to reduce the system to a single agent or a simple MAS, is the most important step compared to the other two in Table 6. This is somewhat unexpected because one may assume that using MAS will always help. We tried the exact setting you mentioned in our preliminary experiments (o3-mini as verifier, GPT-4o as meta-agent), and a stronger verifier indeed gives better results. This matches the oracle experiment in Figure 4B: when the verifier is stronger, the performance of MAS-Zero can further improve. We believe that as stronger LLMs become available, the performance of MAS-Zero will improve as well.

---

> ### Author Response · Authors · 2025-11-18
> **Response to Reviewer gDSq (Part 3)**
>
> > Q3: The system is initialized with four building blocks (CoT, CoT-SC, Debate, Self-Refine). How sensitive is the final performance to this specific set? For instance, what happens if it is only given CoT and Debate? Does the "MAS-Evolve" step successfully reinvent a Self-Refine-like process, or is its creativity constrained to what it is initially given?
>
> The building blocks in MAS-INIT are extensively used in the literature (e.g., [1,2,3]) and already widely accepted and justified. We follow prior work to ensure a fair comparison. The original set of blocks is important because, in MAS-Zero, our goal is not to introduce new agent behaviors or new interaction patterns, but to find a good composition for a given question. We do not expect the MAS-Evolve step to reinvent a Self-Refine-like process; instead, it aims to propose a good composition using the existing blocks.
>
> In L227–L231, we explain that inventing new agent behaviors would require a prohibitively large action space, and our pilot study also showed sub-optimal results in this setting. We believe that a design grounded in the provided building blocks is a more suitable and effective choice.
>
> [1]: Automated design of agentic systems, Hu et al., ICLR 2025.
> [2]: AFlow: Automating agentic workflow generation, Zhang et al., ICLR 2025 Oral.
> [3]: Multi-agent Architecture Search via Agentic Supernet, Zhang et al., ICML 2025 Oral.

---

### Official Review · Reviewer_cRpo · 2025-11-09

**Soundness:** 2
**Presentation:** 2
**Contribution:** 2
**Rating:** 2
**Confidence:** 4

**Summary:**

The work proposes an optimization framework that optimizes a meta-agent which initializes and designs the multi-agent framework. It gathers experiences during the training time and can directly generate well-performing multi-agent during the inference time. It's evaluated on two reasoning tasks, compared against broad baselines, and shown to achieve improved performance and higher cost-efficiency.

**Strengths:**

- Comprehensive baselines.
- Relatively clear presentation.

**Weaknesses:**

- The system seems to need quite some manual design, for example, the modules in the `MAS-INIT`, and prompts in Appendix J. It's not clear how the system is robust w.r.t these manual design and how much replies on human instructions.
- Following the above reason, the novelty might be lacking because the lack of automation.
- Some implementation details to ensure fair comparison is not clear, see questions.
- The title doesn't seem to be precise? Specifically, "ZERO SUPERVISION" holds in the sense that " support adaptivity at inference time, so that MAS designs can betailored per problem instance without relying on training or validation sets. ", but it still requires supervision during meta-agent training.

**Questions:**

- In Table 1, how do you control the input/output token budget to be the same?
- How did you implement the baselines?
- In Figure 1, is it inference cost or optimization cost? If it's one of these two, what's the other?

---

> ### Author Response · Authors · 2025-11-18
> **Response to Reviewer cRpo**
>
> > W1: The system seems to need quite some manual design, for example, the modules in the MAS-INIT, and prompts in Appendix J. It's not clear how the system is robust w.r.t these manual design and how much replies on human instructions. Following the above reason, the novelty might be lacking because the lack of automation.
>
> The building blocks in MAS-INIT are extensively used in the literature (e.g., [1,2,3]) and already widely accepted and justified. Similarly, manually designed prompts is also widely accepted and well supported in the inference-time community ([4,5]). The main contributions of MAS-Zero are: (i) the first inference-time MAS, (ii) SoTA performance, and (iii) key insights from comprehensive experiments. None of these have been achieved in prior work. We believe that the use of these blocks and prompt implementations does not reduce our contributions.
>
> We would also like to note that we use the **same** prompt implementation for the meta-agent across **all** benchmarks (covering a wide range of tasks, including mathematical problems, PhD-level QA, coding, and agentic tasks). The evolution process is entirely automatic. We believe this shows strong automation and generalization in the method, highlighting its broad applicability and robustness.
>
> [1]: Automated design of agentic systems, Hu et al., ICLR 2025
> [2]: AFlow: Automating agentic workflow generation, Zhang et al., ICLR 2025 Oral
> [3]: Multi-agent Architecture Search via Agentic Supernet, Zhang et al., ICML 2025 Oral
> [4]: SELF-DISCOVER: Large Language Models Self-Compose Reasoning Structures, Zhou et al., NeurIPS 2024
> [5]: Gepa: Reflective prompt evolution can outperform reinforcement learning, Agrawal et al., 2025
> > W2: The title doesn't seem to be precise? Specifically, "ZERO SUPERVISION" holds in the sense that " support adaptivity at inference time, so that MAS designs can betailored per problem instance without relying on training or validation sets. ", but it still requires supervision during meta-agent training.
>
> This is likely a misunderstanding. MAS-Zero is an inference-time method and **does not** involve any meta-agent training. Our intention is that MAS-Zero does not rely on external human supervision/label. This usage is not new and also appears in [1]. If this phrasing may introduce confusion, we can consider changing it to “self-supervision” for greater clarity.
>
> [1]: Absolute Zero: Reinforced Self-play Reasoning with Zero Data, Zhao et al., 2025
> > Q1: In Table 1, how do you control the input/output token budget to be the same?
>
> This is likely a misunderstanding. We did not “control” the budget to be the same in Table 1 or Figure 1. We use the official code or model for each baseline and faithfully report the actual budget used in Figure 1, as well as the performance they produce in Table 1.
> > Q2: How did you implement the baselines?
>
> All baselines selected come with official code, which ensures that we follow the exact setup provided by the authors. We only modify the input dataset when the original work did not include the benchmark we consider in MAS-Zero. For the training-based meta-agent baseline, we directly use the trained model provided on Hugging Face.
> > Q3: In Figure 1, is it inference cost or optimization cost? If it's one of these two, what's the other?
>
> In L378–L385, we clarify that the cost “includes both 'training' (if any) and test-time usage.” We believe this is a fair comparison because the central idea of any inference-time method (including MAS-Zero) is to shift the expensive optimization cost into the inference phase (L128–L130). It is expected that such methods will cost more than using **only** the inference phase of an optimization-based method, as the full cost of an optimization-based method includes both optimization and inference. Additionally, optimization-based methods require in-domain validation sets or training sets, which are often unavailable and may not generalize
>
> In the scenario where a user cares strongly about reducing test-time cost, MAS-Zero can disable MAS-Evolve. In this case, instead of running "4 INIT + 5 EVOLVE + 1 JUDGE"," it only needs "4 INIT + 1 JUDGE", which is much cheaper. The table below shows the test-time cost of the optimization-based baselines and MAS-Zero, using GPT-4o as the underlying LLM. We see that while MAS-Zero without MAS-Evolve (MAS-Zero (–MAS-Evolve)) has lower performance than full MAS-Zero, it still outperforms (or matches) the other baselines while using far less cost (smaller than the considered baselines). This demonstrates that MAS-Zero is cost-efficient and can adapt to different scenarios.
> | Method | Inference-time Cost ($) | AIME24 | GPQA |
> | -------- | -------- | -------- | -------- |
> | MaAS     | 18.34     | 12.50     | 43.37 |
> | ADAS     | 6.19     | 0.0     | 45.20 |
> | AFlow     | 5.46     | 20.83     | 46.99 |
> | MAS-Zero     | 15.68     | 33.33     | 50.63 |
> | MAS-Zero (-MAS-Evolve)  | 4.03   | 20.00     | 48.73     |

---

### Comment · Area_Chair_RrtW · 2025-11-27
**Please review the authors' responses and provide feedback ASAP**

Dear Reviewers,

Thank you for your essential contributions to the review process. The authors have submitted their responses to your initial reviews.

I kindly ask you to carefully review the authors' responses for this submission. Your timely assessment of how the authors have addressed your original concerns is a critical step in reaching a final decision.

Please provide your feedback and any necessary updates to your reviews as soon as possible to ensure we can meet our tight schedule for the discussion phase.

Your prompt attention to this matter is highly appreciated.

Regards,

-AC

---

### Note · Authors · 2026-01-06

I have read and agree with the venue's withdrawal policy on behalf of myself and my co-authors.